# Efficient pooling of predictions via kernel embeddings

**Sam Allen**                                                                                              *sam.allen@kit.edu*
*Seminar for Statistics*
*ETH Zürich*

**David Ginsbourger**                                                                        *david.ginsbourger@unibe.ch*
*Institute for Mathematical Statistics and Actuarial Science*
*University of Bern*

**Johanna Ziegel**                                                                            *ziegel@stat.math.ethz.ch*
*Seminar for Statistics*
*ETH Zürich*

**Reviewed on OpenReview:** *https: // openreview. net/ forum? id= dji9MfONNP*

## Abstract

Probabilistic predictions are probability distributions over the set of possible outcomes. Such predictions quantify the uncertainty in the outcome, making them essential for effective decision making. By combining multiple predictions, the information sources used to generate the predictions are pooled, often resulting in a more informative forecast. Probabilistic predictions are typically combined by linearly pooling the individual predictive distributions; this encompasses several ensemble learning techniques, for example. The weights assigned to each prediction can be estimated based on their past performance, allowing more accurate predictions to receive a higher weight. This can be achieved by finding the weights that optimise a proper scoring rule over some training data. By embedding predictions into a Reproducing Kernel Hilbert Space (RKHS), we illustrate that estimating the linear pool weights that optimise kernel-based scoring rules is a convex quadratic optimisation problem. This permits an efficient implementation of the linear pool when optimally combining predictions on arbitrary outcome domains. This result also holds for other combination strategies, and we additionally study a flexible generalisation of the linear pool that overcomes some of its theoretical limitations, whilst allowing an efficient implementation within the RKHS framework. These approaches are compared in an application to operational wind speed forecasts, where this generalisation is found to offer substantial improvements upon the traditional linear pool.

## 1 Introduction

Suppose we are interested in predicting an outcome variable $Y \in \mathcal{Y}$. A point prediction for $Y$ is a single value in $\mathcal{Y}$, whereas a probabilistic prediction generally takes the form of a probability distribution over $\mathcal{Y}$. Probabilistic predictions therefore quantify the uncertainty in the outcome, making them essential for decision making and risk assessment. As such, they have become commonplace when issuing predictions in a variety of application domains, including economics, politics, epidemiology, and the atmospheric sciences.

In practice, we often have access to several predictions, issued by competing models or experts, for example. By combining these predictions, we can leverage the (generally different) information sources used to construct each individual prediction, thereby producing a more informative forecast. Combination strategies, for both point-valued and probabilistic predictions, have received much attention in the forecasting literature (see Winkler et al., 2019; Wang et al., 2023, for recent reviews). The standard approach to combine probabilistic predictions is to *linearly pool* them (Stone, 1961). Linearly pooling predictive distributions is

simple, yet satisfies several appealing properties, and often produces more accurate predictions than more elaborate combination strategies in practical applications (Clemen & Winkler, 1999; Wang et al., 2023).

The linear pool also forms the basis of many *ensemble learning* techniques. Ensemble learning involves training several prediction algorithms, and then combining their output to obtain a prediction that is more robust and accurate than the output of the individual prediction algorithms, or *base learners* (e.g. Dietterich, 2002). The outputs are typically combined by averaging or linearly combining the component predictions. This encompasses bootstrap aggregating (or *bagging*; Breiman, 1996), random forests (Breiman, 2001), and boosting (Freund & Schapire, 1996), for example. Bayesian methods also often involve averaging a large number of candidate models, and hence can be interpreted as a linear pool (Hastie et al., 2017, Chapter 16).

The performance of the linear pool will generally depend on the weights assigned to each component prediction. While it is common to assign equal weight to all predictions, Hall & Mitchell (2007) alternatively suggest linearly pooling probabilistic predictions "by identifying weights that deliver the most 'accurate' density forecast, in a statistical sense." To achieve this, they propose to estimate the weights by optimising a *proper scoring rule* over a set of training data. Proper scoring rules are functions that quantify the accuracy of a probabilistic prediction (see e.g. Gneiting & Raftery, 2007), and optimising a proper scoring rule over a training data set therefore finds the weights that result in the most accurate pooled prediction, with accuracy measured in terms of the chosen scoring rule. Hall & Mitchell (2007) originally used the logarithmic score within this framework, Thorey et al. (2017) instead proposed the continuous ranked probability score (CRPS), while Opschoor et al. (2017) employed weighted versions of these scoring rules that focus on particular outcomes of interest. However, numerical approaches to estimate the weights by optimising a proper scoring rule may be computationally cumbersome.

Several popular scoring rules, including the CRPS, belong to the class of *kernel scores*, scoring rules defined in terms of positive definite kernels (Dawid, 2007; Gneiting & Raftery, 2007). A positive definite kernel on $\mathcal{Y}$ is a symmetric function $k : \mathcal{Y} \times \mathcal{Y} \to \mathbb{R}$ that satisfies

$$\sum_{i=1}^{n} \sum_{j=1}^{n} a_i a_j k(y_i, y_j) \geq 0$$

for any $n \in \mathbb{N}$, $a_1, \ldots, a_n \in \mathbb{R}$, and $y_1, \ldots, y_n \in \mathcal{Y}$. Every positive definite kernel $k$ generates a Hilbert space of functions, referred to as a Reproducing Kernel Hilbert Space (RKHS; Aronszajn, 1950). Kernel methods are well-established in the machine learning literature since positive definite kernels correspond to dot products in (potentially high- or infinite-dimensional) feature spaces, thereby allowing many linear methods in Euclidean space to be generalised to an RKHS (see e.g. Schölkopf et al., 1998). Since probability distributions can be converted to elements in an RKHS via their kernel mean embedding (Muandet et al., 2017), these RKHS methods can also be applied to probabilistic predictions. The connection between kernel methods and proper scoring rules is discussed in detail by Steinwart & Ziegel (2021).

In this paper, we demonstrate that if the weights of the linear pool are estimated by optimising a kernel score over a set of training data, then this is equivalent to a convex quadratic optimisation problem in an RKHS. The optimal weights in the linear pool can therefore be estimated efficiently and robustly. Since kernels, and therefore kernel scores, can be defined on very general outcome spaces, this framework can be applied not only to probabilistic predictions of univariate real-valued outcomes, but also when predicting outcomes in multivariate Euclidean space, spatial-temporal domains, function spaces, graph spaces, and so on. The optimisation problem can be adapted by changing the positive definite kernel, allowing personal preferences and previous results on kernels to be incorporated into the optimisation.

This key result also holds for alternative combination strategies. We study a flexible approach that generalises the linear pool by allowing different component predictions to receive different weight in different regions of the outcome space. This overcomes some well-known limitations of the linear pool when combining probabilistic predictions (Hora, 2004; Gneiting & Ranjan, 2013) by simultaneously re-calibrating the component predictions. When the predictive distributions are discrete, this approach corresponds to a linear pool of the order statistics of the discrete predictive distribution, and can therefore also be implemented efficiently within the RKHS framework. In the following, we study the linear pool applied to predictive distributions, to point predictions, and to order statistics of discrete predictive distributions. The order

statistic-based approach is found to offer substantial improvements upon the traditional linear pool when applied to a case study on operational weather forecasts. The approach generates forecasts that are 10-18% more accurate than when the linear pool weights are estimated by minimising a proper scoring rule, and 20-30% more accurate than when the linear pool weights are assumed equal for all component predictions.

The following section introduces proper scoring rules, and discusses their connection to kernel methods. Section 3 defines the linear pool and illustrates that by embedding probabilistic predictions into an RKHS, estimating the linear pool weights by optimising a kernel score is a convex quadratic optimisation problem. Section 4 discusses the relevance of this approach when combining discrete predictive distributions, and introduces a flexible generalisation of the linear pool that can similarly be implemented efficiently using kernel methods. The practical utility of the proposed approaches is illustrated in an application to weather forecasting in Section 5. Section 6 discusses extensions of the approach to arbitrary outcome domains, before Section 7 concludes. The code and data used in this study are publicly available at `https://github.com/sallen12/RKHSCombi`.

## 2    Forecast evaluation using kernels

Here and throughout the paper, we denote realisations of $Y$ as $y \in \mathcal{Y}$, and use $x$ and $X$ to denote deterministic and random elements in $\mathcal{Y}$.

### 2.1    Proper scoring rules

The accuracy of probabilistic predictions can be quantified using proper scoring rules. A scoring rule is a function $S : \mathcal{F} \times \mathcal{Y} \to \overline{\mathbb{R}}$, where $\mathcal{F}$ is a convex class of probability distributions over $\mathcal{Y}$, and $\overline{\mathbb{R}}$ is the extended real line. Scoring rules assign a numerical score to a prediction and corresponding observation, quantifying the degree of separation between them. The scoring rule $S$ is *proper* (with respect to $\mathcal{F}$) if the expected score is minimised when the true distribution of $Y$ is issued as the prediction. That is, when $Y \sim G$,

$$\mathbb{E}S(G, Y) \leq \mathbb{E}S(F, Y) \tag{1}$$

for all $F, G \in \mathcal{F}$. The scoring rule is *strictly proper* if, in addition, equality in equation 1 holds if and only if $F = G$. We assume throughout that all expectations are finite where necessary.

When evaluating predictive distributions on $\mathcal{Y} \subseteq \mathbb{R}$, a popular scoring rule is the *continuous ranked probability score* (CRPS),

$$\mathrm{CRPS}(F, y) = \mathbb{E}|X - y| - \frac{1}{2}\mathbb{E}|X - X'|, \tag{2}$$

where $X, X' \sim F$ are independent (Matheson & Winkler, 1976; Gneiting & Raftery, 2007). To evaluate multivariate predictions ($\mathcal{Y} \subseteq \mathbb{R}^d$), the CRPS can be generalised to the *energy score*,

$$\mathrm{ES}(F, y) = \mathbb{E}\|X - y\| - \frac{1}{2}\mathbb{E}\|X - X'\|, \tag{3}$$

where $X, X' \sim F$ are independent, and $\|\cdot\|$ is the Euclidean distance in $\mathbb{R}^d$ (Gneiting & Raftery, 2007).

The CRPS and energy score are both examples of kernel scores. The kernel score associated with a positive definite kernel $k$ on $\mathcal{Y}$ is

$$S_k(F, y) = \frac{1}{2}\mathbb{E}k(X, X') + \frac{1}{2}k(y, y) - \mathbb{E}k(X, y), \tag{4}$$

where $X, X' \sim F$ are independent. The energy score is the kernel score associated with the kernel $k(x, x') = \|x\| + \|x'\| - \|x - x'\|$, with the CRPS a particular case when $d = 1$. In the following, we refer to this kernel as the *energy kernel*. The positive definiteness of the energy kernel follows from well-known relationships between positive definite and (conditionally) negative definite kernels (Berg et al., 1984, Lemma 2.1, page 74); see also Sejdinovic et al. (2013, Lemma 12). Not all proper scoring rules are kernel scores — the logarithmic (or negative log-likelihood) score, for example, is not — though the kernel score framework also encompasses many other popular scoring rules, including the Brier score (Brier, 1950), variogram score (Scheuerer &

Hamill, 2015), and the angular CRPS (Grimit et al., 2006). By transforming the kernel $k$, kernel scores can be directed towards particular outcomes, providing a means to construct weighted scoring rules (Allen et al., 2023).

### 2.2 Kernel mean embeddings

In the machine learning literature, it is common to measure the distance between probability distributions using a *maximum mean discrepancy* (MMD), which corresponds to the RKHS distance between the *kernel mean embeddings* of the distributions. If $F$ is a probability distribution on $\mathcal{Y}$, then the kernel mean embedding of $F$ by a positive definite kernel $k$ is

$$\mu_F = \int_{\mathcal{Y}} k(x, \cdot) \, \mathrm{d}F(x),$$

which is a function in the RKHS of $k$, denoted here by $\mathcal{H}_k$. Equivalently, $\mu_F = \mathbb{E}k(X, \cdot)$ with $X \sim F$. Kernel mean embeddings therefore convert probability distributions to elements in an RKHS; if this embedding is injective and the kernel is bounded, then the kernel $k$ is called *characteristic* (Fukumizu et al., 2007). A review of kernel mean embeddings can be found in Muandet et al. (2017).

The MMD between two distributions $F$ and $G$ is then $\|\mu_F - \mu_G\|_{\mathcal{H}_k}$, where $\|\cdot\|_{\mathcal{H}_k}$ is the RKHS norm. In practice, it is common to calculate the squared MMD, which can be expressed as

$$\|\mu_F - \mu_G\|_{\mathcal{H}_k}^2 = \mathbb{E}k(X, X') + \mathbb{E}k(Y, Y') - 2\mathbb{E}k(X, Y), \tag{5}$$

where $X, X' \sim F$ and $Y, Y' \sim G$ are independent. This follows from the reproducing property of an RKHS,

$$k(x, x') = \langle k(x, \cdot), k(x', \cdot) \rangle_{\mathcal{H}_k} \quad \text{for any} \quad x, x' \in \mathcal{Y},$$

where $\langle \cdot, \cdot \rangle_{\mathcal{H}_k}$ is the RKHS inner product (Aronszajn, 1950).

By comparing equations 4 and 5, the kernel score corresponding to $k$ is simply (half) the squared MMD between the probabilistic prediction $F$ and a Dirac measure at the observation $y$, denoted $\delta_y$. The squared MMD at equation 5 is equivalent to the difference in expected scores $\mathbb{E}S_k(F, Y) - \mathbb{E}S_k(G, Y)$, for $Y \sim G$, often called the *score divergence*. The propriety of kernel scores therefore follows from the non-negativeness of the MMD, and, for bounded kernels, a kernel score is strictly proper if and only if $k$ is characteristic, in which case the MMD is a metric (Steinwart & Ziegel, 2021). We leverage these connections between kernel scores and MMDs in the following section to propose an efficient framework with which to optimise kernel scores.

## 3 Efficient linear pooling

Suppose we have access to $J \in \mathbb{N}$ predictive distributions $F_1, \ldots, F_J$ on $\mathcal{Y}$. The standard approach to forecast combination is to *linearly pool* the different predictions:

$$F_{LP} = \sum_{j=1}^{J} w_j F_j, \tag{6}$$

where the weights $w_j \geq 0$ sum to 1 (e.g. Stone, 1961). The weights quantify the relative contributions of the component predictions $F_j$ to the combined prediction $F_{LP}$. The linear pool is simple, interpretable, and performs competitively in many applications; it is therefore widely adopted in practice.

The linear pool prediction also exhibits several desirable theoretical properties. If all component predictions agree on the probability of an event, then the linear pool will also issue this probability. Clemen & Winkler (1999) refer to this as the *unanimity property*. Similarly, if $F_1, \ldots, F_J$ all have the same mean, then this will also be the mean of $F_{LP}$ (e.g. Hall & Mitchell, 2007). Moreover, when $\mathcal{Y} \subseteq \mathbb{R}^d$, for $d > 1$, linearly pooling the marginal distributions of the component predictions is equivalent to taking the marginal distribution of the linear pool prediction; the linear pool is the only combination scheme that satisfies this *marginalisation property* (Genest, 1984).

Lichtendahl Jr. et al. (2013, Proposition 1) additionally show that for a few popular scoring rules, the score assigned to the linear pool prediction is guaranteed to be no larger than the (weighted) average score assigned to the $J$ component predictions. The following proposition generalises their result by showing that this holds for any kernel score.

**Proposition 1.** *Let $S_k$ be a kernel score on $\mathcal{Y}$, and let $F_{LP}$ denote the linear pool at equation 6. Then, for any weights $w_1, \ldots, w_J \geq 0$ that sum to 1, and any $y \in \mathcal{Y}$,*

$$S_k(F_{LP}, y) \leq \sum_{j=1}^{J} w_j S_k(F_j, y). \tag{7}$$

*Proof.* Clearly, equation 7 holds more generally for any scoring rule that is convex in the forecast argument. Convexity of kernel scores follows from linearity of kernel mean embeddings and convexity of squared norms. Let $F_1, F_2$ be predictive distributions on $\mathcal{Y}$, and, given a positive definite kernel $k$ on $\mathcal{Y}$, let $\mu_{F_1} = \mathbb{E}k(X_1, \cdot)$ and $\mu_{F_2} = \mathbb{E}k(X_2, \cdot)$ be the kernel mean embeddings of $F_1$ and $F_2$ into $\mathcal{H}_k$, where $X_1 \sim F_1$, and $X_2 \sim F_2$. An observation $y \in \mathcal{Y}$ corresponds to the function $\mu_{\delta_y} = k(y, \cdot)$ in the RKHS. For any $\lambda \in [0, 1]$, we have

$$\begin{aligned}
S_k(\lambda F_1 + (1 - \lambda)F_2, y) &= \frac{1}{2}\|\lambda \mu_{F_1} + (1 - \lambda)\mu_{F_2} - \mu_{\delta_y}\|_{\mathcal{H}_k}^2 \\
&= \frac{1}{2}\|\lambda(\mu_{F_1} - \mu_{\delta_y}) + (1 - \lambda)(\mu_{F_2} - \mu_{\delta_y})\|_{\mathcal{H}_k}^2 \\
&\leq \frac{\lambda}{2}\|\mu_{F_1} - \mu_{\delta_y}\|_{\mathcal{H}_k}^2 + \frac{(1 - \lambda)}{2}\|\mu_{F_2} - \mu_{\delta_y}\|_{\mathcal{H}_k}^2 \\
&= \lambda S_k(F_1, y) + (1 - \lambda)S_k(F_2, y).
\end{aligned}$$

$\square$

The difference between the left- and right-hand sides of equation 7 is given in independent work by Krüger (2024, Proposition 5.1). An open question is how this difference is affected by properties of the kernel (and more generally the choice of scoring rule).

The performance of the linear pool will also depend on the weights in equation 6. Hall & Mitchell (2007) propose finding the weights that optimise a proper scoring rule over a set of training data. In practice, we observe a sequence of $n$ observations $y_i \in \mathcal{Y}$, and have access to the corresponding predictions $F_{1,i}, \ldots, F_{J,i}$, for $i = 1, \ldots, n$. We then want to find the weights $\mathbf{w} = (w_1, \ldots, w_J)^\top \in [0, 1]^J$ in equation 6 that optimise the empirical score of the linear pool prediction over these $n$ observations,

$$\hat{S}_n(\mathbf{w}) = \sum_{i=1}^{n} \alpha_i S \left( \sum_{j=1}^{J} w_j F_{j,i}, y_i \right), \tag{8}$$

where $S$ is a proper scoring rule, and $\sum_{j=1}^{J} w_j = 1$. The scaling parameters $\alpha_1, \ldots, \alpha_n \geq 0$ are often all set to 1, but can also be used to emphasise particular prediction-observation pairs: in sequential prediction settings, for example, it is common to assign higher weight to more recently collected data.

Hall & Mitchell (2007) originally employed the logarithmic score within equation 8, which corresponds to maximum likelihood estimation. Several recent studies have instead proposed estimating the weights of the linear pool by minimising the empirical CRPS (see Berrisch & Ziel, 2023, and references therein). The CRPS at equation 2 is more commonly written as

$$\mathrm{CRPS}(F, y) = \int_{\mathbb{R}} (F(z) - \mathbb{1}\{y \leq z\})^2 \, \mathrm{d}z,$$

where $\mathbb{1}\{\cdot\}$ is the indicator function, and $F$ is the predictive distribution function (Matheson & Winkler, 1976). However, from this representation, it is not immediately obvious how the empirical score at equation 8 depends on the weights of the linear pool. Leveraging the kernel score representation of the CRPS at

equation 2, the following proposition shows that the CRPS is a quadratic function of the linear pool weights, allowing them to be estimated robustly and efficiently. More generally, Proposition 2 shows that if $S$ is a kernel score, then estimating the weights in the linear pool that optimise the empirical score at equation 8 is a convex quadratic optimisation problem.

**Proposition 2.** *For a kernel score $S_k$ on $\mathcal{Y}$, the mapping $\mathbf{w} \mapsto \hat{S}_n(\mathbf{w})$ is quadratic. In particular, finding the weights in the linear pool that optimise a kernel score is a convex quadratic programming problem.*

*Proof.* For $i = 1, \ldots, n$, let $X_{j,i} \sim F_{j,i}$, let $F_{LP,i} = \sum_{j=1}^{J} w_j F_{j,i}$, and let

$$\mu_{F_{LP,i}} = \sum_{j=1}^{J} w_j \mathbb{E}k(X_{j,i}, \cdot)$$

denote the kernel mean embedding of the linear pool prediction $F_{LP,i}$. Then, $\hat{S}_n(\mathbf{w})$ becomes

$$\sum_{i=1}^{n} \alpha_i S_k(F_{LP,i}, y_i) = \frac{1}{2} \sum_{i=1}^{n} \alpha_i \|\mu_{F_{LP,i}} - \mu_{\delta_{y_i}}\|_{\mathcal{H}_k}^2. \tag{9}$$

Since $\|h\|_{\mathcal{H}_k}^2 = \langle h, h \rangle_{\mathcal{H}_k}$ for any $h \in \mathcal{H}_k$, using the linearity of inner products and the reproducing property of an RKHS, minimising 9 is equivalent to minimising

$$\sum_{i=1}^{n} \alpha_i \left[ \sum_{j=1}^{J} \sum_{l=1}^{J} w_j w_l \mathbb{E}k(X_{j,i}, X_{l,i}) - 2 \sum_{j=1}^{J} w_j \mathbb{E}k(X_{j,i}, y_i) \right].$$

Define $A \in \mathbb{R}^{J \times J}$ such that

$$A_{j,l} = \sum_{i=1}^{n} \alpha_i \mathbb{E}k(X_{j,i}, X_{l,i}) \quad \text{for } j, l = 1, \ldots, J, \tag{10}$$

and define $\mathbf{c} \in \mathbb{R}^J$ such that

$$c_j = -\sum_{i=1}^{n} \alpha_i \mathbb{E}k(X_{j,i}, y_i) \quad \text{for } j = 1, \ldots, J. \tag{11}$$

Then, equation 9 can be written in matrix notation as $\mathbf{w}^\top A \mathbf{w} + 2\mathbf{c}^\top \mathbf{w}$.

Hence, optimising the weights in the linear pool is a convex quadratic optimisation problem of the form

$$\text{minimise} \quad \frac{1}{2} \mathbf{w}^\top A \mathbf{w} + \mathbf{c}^\top \mathbf{w} \tag{12}$$

subject to $\mathbf{1}^\top \mathbf{w} = 1$ and $0 \preceq \mathbf{w}$, that is, $w_j \geq 0$ for $j = 1, \ldots, J$. $\qquad \square$

**Remark 3.** When $S$ is a kernel score, estimating the linear pool weights that minimise the empirical score of equation 8 boils down to optimising a non-negative linear combination of maximum mean discrepancies. Proposition 2 therefore illustrates how the convex quadratic structure of well-established optimisation problems from the realm of kernel methods, such as kernel ridge regression, also arises in the context of probabilistic forecasting. The proposition could also be extended further by adding a regularisation term, akin to kernel ridge regression.

By employing the energy kernel (with $d = 1$) in the framework of Proposition 2, we can efficiently estimate the weights in the linear pool that optimise the CRPS. But Proposition 2 holds more generally for any positive definite kernel on $\mathcal{Y}$, and the kernel could also be chosen to emphasise particular aspects of the predictive distribution. Opschoor et al. (2017) proposed estimating the linear pool weights by optimising a weighted version of the CRPS that assigns more weight to particular outcomes (see Gneiting & Ranjan, 2011). Allen

et al. (2023) demonstrate that popular weighted versions of kernel scores are generally themselves kernel scores, so this approach also falls into the scope of Proposition 2. This provides an example of how the kernel can be chosen to incorporate personal beliefs and information into the optimisation problem.

While linear pooling is most commonly studied when $\mathcal{Y} \subseteq \mathbb{R}$, Proposition 2 makes no assumptions on the outcome space except that a positive definite kernel exists on this domain. This provides the potential for efficient forecast combination strategies on arbitrary outcome domains. For example, the weights in a multivariate linear pool could be estimated by employing the energy kernel (with $d > 1$) in the framework of Proposition 2, which is equivalent to optimising the average energy score over the training data. Well known results on kernels could also be leveraged to linearly pool predictions made on other domains, including function spaces (Wynne & Duncan, 2022), graph spaces (Kriege et al., 2020), and Banach spaces (Ziegel et al., 2024).

Regardless of the domain, by converting optimum score estimation to a convex quadratic optimisation problem, it can be performed efficiently using existing programming software; in the following, we use the `kernlab` package in `R` (Karatzoglou et al., 2004).

## 4 Efficient pooling of discrete predictive distributions

### 4.1 Linearly pooling discrete predictive distributions

In theory, Proposition 2 holds regardless of the form of the component predictive distributions. In practice, we require the kernel mean embeddings associated with each prediction in order to construct the kernel matrix $A$ and the vector $\mathbf{c}$ in equation 12. In Appendix A, we provide the kernel mean embeddings for univariate Gaussian component distributions under the energy kernel. While this may not be possible for certain kernels and parametric families of distributions, it becomes trivial when the predictions $F_{1,i}, \ldots, F_{J,i}$ take the form of a finite sample, as is commonly the case in practice. We refer to such predictions as *discrete predictive distributions*. Discrete predictive distributions are also often referred to as ensemble forecasts (e.g. Leutbecher & Palmer, 2008), but we avoid this terminology to avoid confusion with ensemble learning methods.

A discrete predictive distribution can be interpreted as a sample of point predictions, $x^1, \ldots, x^M \in \mathcal{Y}$, with the empirical distribution of the sample defining a probabilistic prediction. Discrete predictive distributions can be obtained, for example, from Markov chain Monte Carlo (MCMC) output, generative machine learning models, Bayesian neural networks, or by aggregating point predictions generated by several competing methods. They are ubiquitous in high-dimensional prediction settings, where it is difficult to specify a complete predictive distribution, and in domains where predictions are generated by multiple runs of computationally expensive physical models, such as in weather and climate forecasting and oceanography.

Suppose that $F_{j,i}$ is a discrete predictive distribution comprised of $M_j$ sample members, $x^1_{j,i}, \ldots, x^{M_j}_{j,i} \in \mathcal{Y}$, for $i = 1, \ldots, n$, $j = 1, \ldots, J$, where the subscript $j$ refers to the component prediction, $i$ indexes over the training data set, and the superscript denotes the member of the sample. That is,

$$F_{j,i} = \frac{1}{M_j} \sum_{m=1}^{M_j} \delta_{x^m_{j,i}}. \tag{13}$$

Given a positive definite kernel $k$ on $\mathcal{Y}$, the kernel mean embedding of the discrete predictive distribution $F_{j,i}$ is simply $\mu_{F_{j,i}} = \sum_{m=1}^{M_j} k(x^m_{j,i}, \cdot)/M_j$. Hence, by Proposition 2, linearly combining discrete predictive distributions can be performed efficiently by computing the kernel matrix $A$ in equation 10 and the vector $\mathbf{c}$ in equation 11, and solving the associated quadratic optimisation problem at equation 12.

**Example 4** (Linearly pooling discrete predictive distributions). The traditional linear pool of the $J$ discrete predictive distributions $F_{j,i}$ is the predictive distribution

$$F_{LP,i} = \sum_{j=1}^{J} \frac{w_j}{M_j} \sum_{m=1}^{M_j} \delta_{x^m_{j,i}},$$

for $i = 1, \ldots, n$. In this case, the kernel matrix in equation 10 has entries

$$A_{j,l} = \sum_{i=1}^{n} \frac{\alpha_i}{M_j M_l} \sum_{m=1}^{M_j} \sum_{m'=1}^{M_l} k(x_{j,i}^m, x_{l,i}^{m'}) \quad \text{for } j, l = 1, \ldots, J,$$

with the vector $\mathbf{c}$ such that

$$c_j = -\sum_{i=1}^{n} \frac{\alpha_i}{M_j} \sum_{m=1}^{M_j} k(x_{j,i}^m, y_i) \quad \text{for } j = 1, \ldots, J.$$

These terms can easily be calculated for any discrete predictive distribution and positive definite kernel. Plugging them into equation 12 then results in an efficient approach to estimate the optimal weights assigned to each component prediction.

In equation 13, it is assumed that each member of the sample in the discrete predictive distribution is equally weighted. However, if the members are not exchangeable, then a different weight could also be assigned to different members of the sample. These weights should reflect the relative performance of each sample member, and can again be estimated by optimising a proper scoring rule over a set of training data. This provides a means to obtain a probabilistic prediction by combining point predictions.

**Example 5** (Linearly pooling point predictions)**.** Assigning weight to each member of the sample yields the predictive distribution

$$F_{SM,i} = \sum_{j=1}^{J} \sum_{m=1}^{M_j} w_{j,m} \delta_{x_{j,i}^m}, \tag{14}$$

for $i = 1, \ldots, n$, where the weights $w_{j,m} \geq 0$ are such that $\sum_{j=1}^{J} \sum_{m=1}^{M_j} w_{j,m} = 1$. The relative contribution of each prediction $F_j$ can be quantified using the cumulative weight assigned to the members of this sample, $\sum_{m=1}^{M_j} w_{j,m}$. This nests the linear pool as a particular case when $w_{j,1} = \cdots = w_{j,M_j}$ for all $j = 1, \ldots, J$.

This approach is equivalent to treating each member of the sample as an individual point prediction, and linearly pooling these $M = M_1 + \cdots + M_J$ predictions. In particular, equation 14 can alternatively be written as

$$F_{SM,i} = \sum_{j=1}^{M} \tilde{w}_j \delta_{\tilde{x}_{j,i}},$$

where $\tilde{x}_{j,i}$ is the $j$-th element in $x_{1,i}^1, \ldots, x_{1,i}^{M_1}, x_{2,i}^1, \ldots, x_{2,i}^{M_2}, \ldots, x_{J,i}^1, \ldots, x_{J,i}^{M_J}$, and $\tilde{w}_j$ is similarly the $j$-th element in $w_{1,1}, \ldots, w_{1,M_1}, \ldots, w_{J,1}, \ldots, w_{J,M_J}$. In this case, the kernel matrix in equation 10 can be written as a matrix of dimension $M \times M$ with entries $A_{j,l} = \sum_{i=1}^{n} \alpha_i k(\tilde{x}_{j,i}, \tilde{x}_{l,i})$ for $j, l = 1, \ldots, M$, while the vector $\mathbf{c}$ contains the values $c_j = -\sum_{i=1}^{n} \alpha_i k(\tilde{x}_{j,i}, y_i)$ for $j = 1, \ldots, M$, and similarly has length $M$. Plugging these into equation 12 returns the optimum weight assigned to each individual sample member.

While equation 14 is a linear combination of the individual sample members, it is not a linear combination of the discrete predictive distributions; by weighting the sample members, the shape of the predictive distributions can change. Hence, the RKHS framework of Proposition 2 can also be used to perform efficient non-linear combinations of the predictive distributions.

## 4.2 Linearly pooling order statistics

By assigning weight to different sample members, Example 5 implicitly assumes that the different sample members (or the processes underlying them) exhibit different predictive performance. If the sample members are exchangeable, then the weights assigned to them should be equal. In this case, the resulting combination in Example 5 reverts to the linear pool of the component predictions. As an alternative, when $\mathcal{Y} \subseteq \mathbb{R}$, we propose assigning different weights to the order statistics of the discrete predictive distributions.

**Example 6** (Linearly pooling order statistics). Let $\mathcal{Y} \subseteq \mathbb{R}$. For $i = 1, \ldots, n$, $j = 1, \ldots, J$, and $m = 1, \ldots, M_j$, let $x_{j,i}^{(m)}$ denote the $m$-th order statistic of the discrete predictive distribution $x_{j,i}^1, \ldots, x_{j,i}^{M_j}$. That is, $x_{j,i}^{(1)} \leq x_{j,i}^{(2)} \leq \cdots \leq x_{j,i}^{(M_j)}$ for all $i, j$. Consider the predictive distribution

$$F_{OS,i} = \sum_{j=1}^{J} \sum_{m=1}^{M_j} w_{j,m} \delta_{x_{j,i}^{(m)}}, \tag{15}$$

where the weights $w_{j,m} \geq 0$ are such that $\sum_{j=1}^{J} \sum_{m=1}^{M_j} w_{j,m} = 1$.

For fixed $i$, this approach resembles equation 14, in that it constitutes a weighting of each sample member, and the relative contribution of the discrete predictive distribution $F_j$ to the combination can again be quantified using $\sum_{m=1}^{M_j} w_{j,m}$. However, the weights in equation 15 are estimated for each order statistic, rather than for each individual sample member. Since it is generally unlikely that the same sample member will always correspond to the same order statistic (especially if the members are exchangeable), this can yield very different predictive distributions. In particular, equation 15 treats the sample as a probabilistic predictive distribution, rather than treating each member of the sample as an individual point prediction. The weights can then transform the predictive distributions, providing a means to re-calibrate the pooled distribution.

As before, equation 15 is a non-linear combination of the discrete predictive distributions, but a linear combination of the individual sample members. Hence, the kernel matrix $A$ and the vector $\mathbf{c}$ can be constructed similarly to as in Example 5, but with $x_{j,i}^m$ replaced with $x_{j,i}^{(m)}$, and this can therefore also be implemented efficiently using the results from Proposition 2.

**Remark 7.** When $\mathcal{Y} \subseteq \mathbb{R}$ and the component distributions $F_j$ have corresponding quantile function $F_j^{-1}$, Berrisch & Ziel (2023) propose an adaptive quantile averaging-based combination scheme:

$$F_{QA}^{-1}(\alpha) = \sum_{j=1}^{J} w_j(\alpha) F_j^{-1}(\alpha), \quad \alpha \in (0, 1). \tag{16}$$

This approach uses a weight function that adapts depending on the quantile level $\alpha$. While we thus far treat the $x_{j,i}^m$ as samples from an underlying distribution, they could also be interpreted as predictive quantiles: if $M_1 = \cdots = M_J$, then the order statistics of the discrete predictive distribution $F_j$ would correspond to the quantiles $F_j^{-1}(\alpha)$ at fixed quantile levels $\alpha$. In this case, estimating the weights $w_{j,1}, \ldots, w_{j,M_j}$ as in Example 6 would return the weight functions $w_j$ in equation 16. Hence, when working with discrete predictive distributions, Proposition 2 can be leveraged to efficiently estimate the weights in equation 16 that are optimal with respect to the CRPS (or any other kernel score).

More generally, given $J$ predictive distributions $F_1, \ldots, F_J$ (not necessarily discrete), and dropping the subscript $i$ for ease of presentation, the approach in Example 6 corresponds to the combination strategy

$$F_{OS}(B) = \sum_{j=1}^{J} \int_B w_j \, \mathrm{d}F_j, \quad B \subseteq \mathcal{Y}, \tag{17}$$

where $w_j : \mathcal{Y} \to \mathbb{R}_{\geq 0}$ are non-negative weight functions, with the requirement that $\sum_{j=1}^{J} \int_{\mathcal{Y}} w_j \, \mathrm{d}F_j = 1$. The relative contribution of each component distribution to the combination can be quantified using $\int_{\mathcal{Y}} w_j \, \mathrm{d}F_j$. If the predictive distributions $F_j$ all have a continuous density function $f_j$, then equation 17 is equivalent to a point-wise linear combination of the component densities. That is,

$$f_{OS}(x) = \sum_{j=1}^{J} w_j(x) f_j(x), \quad x \in \mathcal{Y}, \tag{18}$$

where $f_{OS}$ is the density function associated with $F_{OS}$. Hence, while Example 6 restricts attention to $\mathcal{Y} \subseteq \mathbb{R}$, in theory, this combination formula can be applied on any domain. The weight function can then be

interpreted as the weight assigned to each component prediction at each point in outcome space, recognising that different predictions may perform better or worse at forecasting particular outcomes.

The linear pool is again recovered when all weight functions in equation 17 are constant, and this combination strategy therefore provides a flexible generalisation of the traditional linear pool. Importantly, this generalisation overcomes some theoretical limitations of the traditional linear pool in equation 6: when $\mathcal{Y} \subseteq \mathbb{R}$, Hora (2004) demonstrates that if the component predictions $F_j$ are all calibrated, in the sense of established notions of probabilistic forecast calibration, then the linear pool will necessarily be over-dispersive and therefore miscalibrated (see also Gneiting & Ranjan, 2013, Theorem 3.1). This trivially extends to the marginals of the linear pool prediction when $\mathcal{Y} \subseteq \mathbb{R}^d$.

On the other hand, if the component predictions are all calibrated, then it is also possible that the combined prediction in equation 17 is calibrated. In particular, the following proposition shows that for continuous predictive distributions, the combination formula in equation 17 satisfies a desirable universality property, in that it can reproduce any continuous probability distribution whose support is the union of the supports of the component distributions. One immediate consequence of this result is that the combination strategy is *(exchangeably) flexibly dispersive* in the sense of Gneiting & Ranjan (2013), while the linear pool is not. In essence, the weighting transforms the predictive distributions, and therefore simultaneously re-calibrates the pooled distributions when estimating the weights. The shape of the weight function allows for a better understanding of how the predictions are transformed, and in which regions of the outcome space each component prediction is valuable.

**Proposition 8.** *Let $\mathcal{M}_{\mathcal{Y}}$ be a class of absolutely continuous probability distributions on $\mathcal{Y}$, and consider any $F_0, F_1, \ldots, F_J \in \mathcal{M}_{\mathcal{Y}}$, with corresponding densities $f_0, f_1, \ldots, f_J$, such that*

$$\mathrm{supp}(F_0) = \mathrm{supp}(F_1) \cup \cdots \cup \mathrm{supp}(F_J).$$

*Then, there exist non-negative weight functions $w_1, \ldots, w_J$ such that*

$$\sum_{j=1}^{J} w_j(x) f_j(x) = f_0(x) \quad \text{for all } x \in \mathcal{Y}.$$

*Proof.* Let $f_+ = \sum_{j=1}^{J} f_j$, and let $w_j(x) = f_0(x)/f_+(x)$ for any $x \in \mathrm{supp}(F_0)$ and 0 otherwise, for all $j = 1, \ldots, J$. Then, for any $x \in \mathcal{Y}$,

$$\sum_{j=1}^{J} w_j(x) f_j(x) = w_1(x) f_+(x) = f_0(x).$$

$\square$

An analogous result holds when $F_0, \ldots, F_J$ are not continuous but have finite support.

**Remark 9.** The proof of Proposition 8 does not require that the weight functions $w_1, \ldots, w_J$ are different. Hence, even if we only have one (absolutely continuous) predictive distribution $F_1$ that has the same support as $F_0$, then there exists a weight function $w_1$ such that $w_1(x) f_1(x) = f_0(x)$ for all $x \in \mathcal{Y}$. This weighting therefore additionally provides a flexible method to re-calibrate or post-process predictive densities. However, estimating this weight function generally does not constitute a convex quadratic optimisation problem. To see this, consider the more general case where, given a positive definite kernel $k$, we wish to find the weight functions $w_1, \ldots, w_J$ in equation 18 that optimise the corresponding empirical kernel score. As in the proof of Proposition 2, by embedding the predictions into an RKHS, the weight functions can be estimated by minimising

$$\sum_{i=1}^{n} \alpha_i \left[ \sum_{j=1}^{J} \sum_{l=1}^{J} \mathbb{E}\left[k(X_{j,i}, X_{l,i}) w_j(X_{j,i}) w_l(X_{l,i})\right] - 2 \sum_{j=1}^{J} \mathbb{E}\left[k(X_{j,i}, y_i) w_j(X_{j,i})\right] \right].$$

This does not fall under the framework of Proposition 2 in full generality. However, this can be circumvented by defining the weight functions $w_1, \ldots, w_J$ as linear combinations of basis functions, in which case the

average kernel score becomes linear in the basis functions' coefficients. This is discussed in detail in Appendix B. In the following section, we consider an application of different pooling methods to discrete predictive distributions, and therefore leave an application of this continuous weight estimation for future work.

## 5 Application to weather forecasts

Consider an application of the combination schemes introduced in the previous sections in the context of weather forecasting. Weather forecasts typically take the form of discrete predictive distributions (more commonly referred to as ensemble forecasts), generated from various numerical or AI-based weather prediction models. We consider the predictions issued by three operational forecast models at the Swiss Federal Office for Meteorology and Climatology (MeteoSwiss). We pool the forecasts from the three models into one predictive distribution, and use this to estimate the relative importance of each model. For each forecast combination strategy, we study how the weights assigned to the different models vary in time, space, and for different forecast horizons. The forecast models all issue predictions in the form of discrete predictive distributions, and they can therefore be pooled efficiently by leveraging the results in the previous sections.

### 5.1 Data

We consider wind speed forecasts issued at 00 UTC for hourly intervals from 1 to 33 hours in advance. For this time range, MeteoSwiss have 3 forecast models in operation:

- **COSMO-1E**: 11 sample members run at a finer spatial resolution;

- **COSMO-2E**: 21 sample members run at medium spatial resolution;

- **ECMWF IFS**: 51 sample members run at a coarser spatial resolution.

The sample members of all models are generated from numerical weather prediction models that involve integrating an initial state of the atmosphere through time according to physical laws. All discrete predictive distributions include a *control member*, which corresponds to the weather model run from a best guess of the initial state of the atmosphere, with the remaining sample members generated by randomly perturbing this best guess, and running the weather model from these perturbed initial conditions. Further details about the models can be found in Keller et al. (2021) and references therein, while Buizza (2018) provides a more comprehensive overview of numerical weather prediction.

The weather model output has been interpolated to 82 weather stations across Switzerland, shown in Figure 1 as a function of altitude. Switzerland has a markedly varied topography, and the altitudes of the stations range from 203m to 3130m above sea level. Forecasts are available daily for three years between the 2nd June 2020 and 31st May 2023, during which none of the three models undergo major changes. Some dates are missing for all models and stations, resulting in a total of $1030 \times 82 = 84460$ forecast-observation pairs for each of the three models.

Rather than issuing a wind speed forecast obtained from only one of the models, operational weather centres wish to combine the information within their various models to yield a more informative prediction. We illustrate how this can be achieved using the framework discussed in the previous sections. We compare the forecasts for several competing approaches: 1) the discrete predictive distributions issued by each of the three individual weather models (*COSMO-1E*, *COSMO-2E*, and *ECMWF IFS*); 2) a linear pool prediction that assigns equal weight to the three forecast models, often referred to as a multi-model forecast in the weather forecasting literature (*LP Equal*); 3) a traditional linear pool of the three discrete predictive distributions with weights estimated by minimising a proper scoring rule, as in Example 4 (*LP Discrete*); 4) a linear pool of the point predictions obtained from the sample members of the three forecast models, as in Example 5 (*LP Point*); and 5) a linear pool of the order statistics of the three forecast models, as in Example 6 (*LP Ordered*).

For the latter three methods, the weights are estimated by minimising the CRPS over a training data set, performed using the energy kernel in the convex quadratic optimisation problem outlined in Proposition 2.

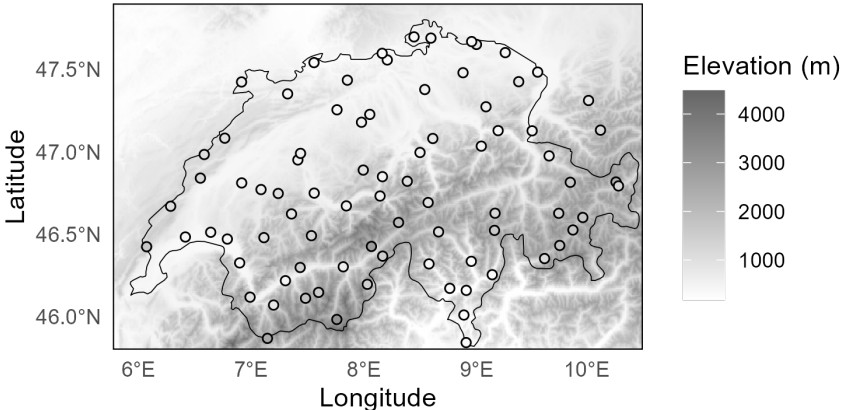

Figure 1: The 82 weather stations in Switzerland.

The CRPS is the canonical scoring rule used to evaluate meteorological forecasts, and therefore also the canonical scoring rule with which to perform model training. Analogous conclusions are drawn when the energy kernel is replaced with other popular kernels, such as the Gaussian, Laplace, and Matern kernels, results for which are shown in Section 5.4. The first two years of data (2nd June 2020 to 1st June 2022) are used as training data on which to estimate the weights, and the resulting forecasts are then assessed out-of-sample using the remaining year of data.

The weights of the combination strategies are first estimated at each station separately. Estimating weights separately at each station allows spatial patterns in the weights to be studied, but ignores the multivariate structure of the forecasts. For example, one forecast could be accurate at the individual stations, but could completely misrepresent the dependence between different stations. Hence, this forecast would receive a lower weight if the multivariate forecasts were combined, where each station corresponds to a different dimension. This is discussed further in Section 6.

### 5.2 Patterns in sample member importance

Firstly, we wish to investigate how the relative importance of the different sample members and discrete predictive distributions evolves over time and space. This can be achieved as in Example 5. Figure 2 shows the average weights assigned to each sample member as a function of the mean squared error (MSE) of the corresponding point prediction. Sample members obtained from the same weather models receive a similar weight, with the higher resolution COSMO-1E model typically receiving a higher weight than the lower resolution models, which aligns with these predictions receiving a lower MSE. Interestingly, the control member of the discrete predictive distributions typically results in the best point-valued prediction of the observation, but receives the least weight when combining the sample members into a probabilistic prediction. This could be because the uncertainty in the sample members is more useful when generating a predictive distribution, rendering the control member less informative.

The weights assigned to the different sample members can be summed to get the overall contribution from each discrete predictive distribution. These weights are shown as a function of the forecast horizon, or forecast lead time, in Figure 3. Since the forecasts are initialised at 00 UTC, a lead time of 1 hour corresponds to 01 UTC, and so on. COSMO-1E receives the highest weight at all lead times, followed by COSMO-2E and then ECMWF IFS. The relative importance of the forecast models changes slightly throughout the day, with COSMO-1E particularly important during the early afternoon, when wind speeds are generally at their peak. The results are analogous if the weight of each model is obtained using the traditional linear pool.

Figure 4 displays the forecast model that receives the highest weight (on average) at each spatial location. The ECMWF IFS model tends to receive a lower weight at weather stations in valleys and on mountain tops, whereas the COSMO-1E typically performs best at these locations. This is likely because the low resolution

model does not accurately capture changes in the local topography and therefore exhibits a larger bias when compared to the observations at the mountainous stations.

Figure 5 shows the CRPS for the various models as a function of the forecast lead time, aggregated across time and station. The ordering of the forecast models is generally insensitive to the lead time: the ECMWF IFS model results in the least accurate forecasts, followed by the COSMO-2E and then the COSMO-1E models. An equal-weighted linear pool outperforms the three individual models at some lead times, but also generates less accurate forecasts than the COSMO-1E at other lead times. Linear pooling the three forecast models, with weights estimated by minimising the CRPS using the kernel framework, systematically outperforms all of the individual models, resulting in CRPS values that are 10-16% lower than those of the COSMO-1E forecasts. By estimating separate weights for each sample member of the forecast models, predictions can be obtained that are 3-6% more accurate than those generated using the traditional linear pool.

### 5.3 Relative order statistic importance

The results shown thus far demonstrate how the relative importance of the different weather models, and each of their sample members, can be determined using kernel mean embeddings, and that we can then investigate how these weights change over time, space, and forecast horizon. However, as discussed in Example 6, weights could alternatively be estimated for order statistics of the discrete predictive distributions, rather than the individual sample members. This corresponds to a flexible generalisation of the linear pool.

The raw wind speed forecasts generated from each of the three weather models are systematically miscalibrated. In particular, they exhibit a severe positive bias, with predicted wind speeds typically larger than those observed, and are also under-dispersive (see probability integral transform (PIT) histograms in Figure 16 in the appendix). It is well-established that numerical weather models generally exhibit large biases when predicting surface weather variables, and fail to accurately quantify the uncertainty in the outcome. By assigning and estimating weights corresponding to order statistics of the discrete predictive distributions, we essentially transform the predictive distributions, thereby re-calibrating the predictions whilst simultaneously quantifying their relative importance.

Figure 6 shows the average weight assigned to each order statistic of the three discrete predictive distributions. The weights are again obtained by using the energy kernel within the quadratic optimisation problem of Proposition 2. In light of the biases in the weather models, the weight profile in Figure 6 demonstrates that a large weight is assigned to the smallest and largest members of the discrete predictive distributions, while all other sample members receive a comparatively low weight. By assigning a large weight to the first order

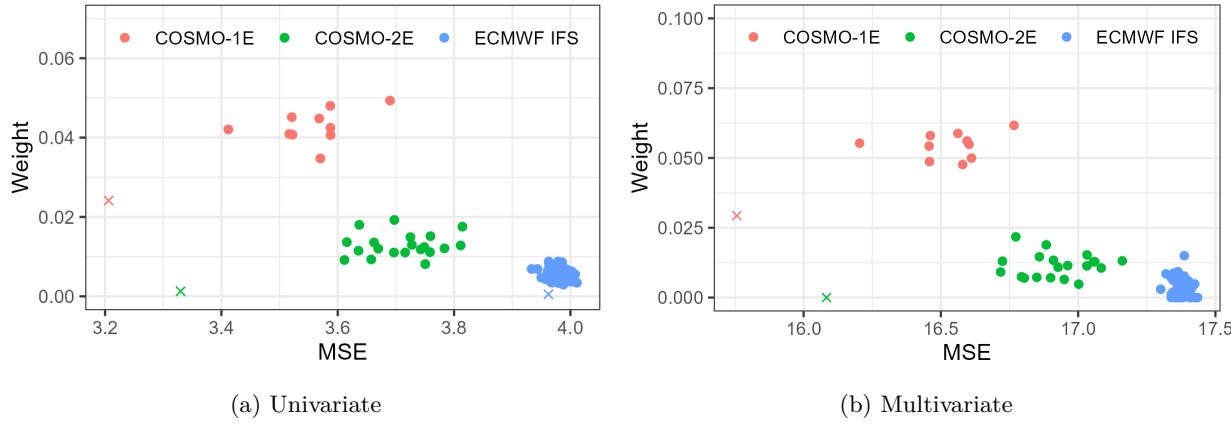

(a) Univariate            (b) Multivariate

Figure 2: The average weight assigned to each sample member against the member's mean squared error. The × symbol corresponds to the control member of each discrete predictive distribution. Results are shown at a lead time of 18 hours. In the univariate case, the weights are averaged across all stations.

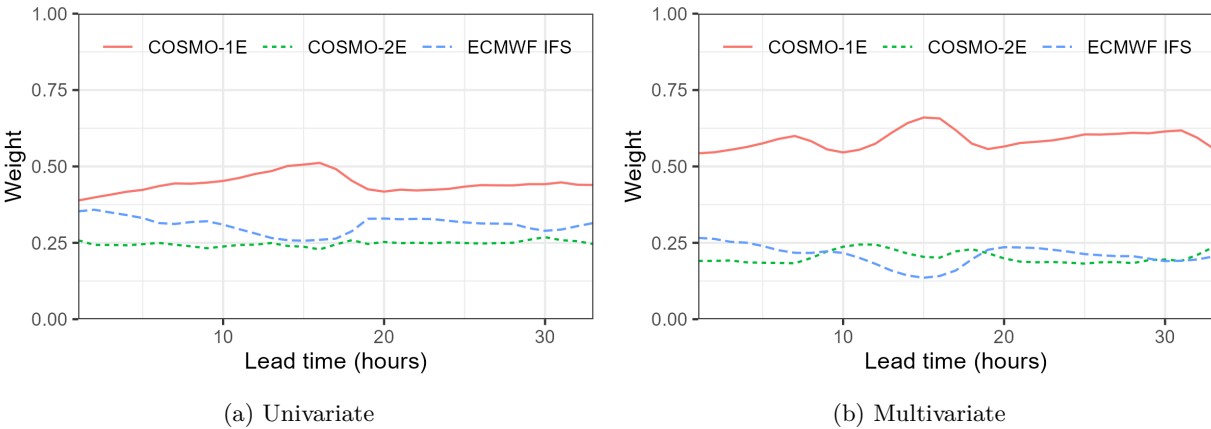

(a) Univariate      (b) Multivariate

Figure 3: The average weight assigned to each forecast model as a function of lead time. In the univariate case, the weights are averaged across all stations.

statistic, the distribution function is shifted towards lower values, helping to account for the positive bias in the predictions. A large weight for the last order statistic then ensures that the spread of the distribution is large enough to quantify the uncertainty in the predictions, helping to account for the under-dispersion of the predictive distributions.

Weighting the order statistics of the discrete predictive distributions yields large improvements upon all other methods: the resulting predictions are consistently 20-30% more accurate than the multi-model forecasts, when assessed using the CRPS, and are between 10% and 18% more accurate than those generated using the traditional linear pool. The average accuracy of the four linear pool-based combination methods, as assessed using the CRPS, is displayed in Figure 7.

This order statistics-based approach performs particularly well in this application because the original component predictions are strongly biased, and, in contrast to the traditional linear pool, it allows the predictive distributions to be transformed using a weight function, helping to alleviate these biases. However, the approach also performs well when the predictions are not biased. In practice, numerical weather forecasts typically undergo some form of statistical post-processing to re-calibrate the weather model output. In Appendix C, we present the same results for the different combination schemes proposed herein applied

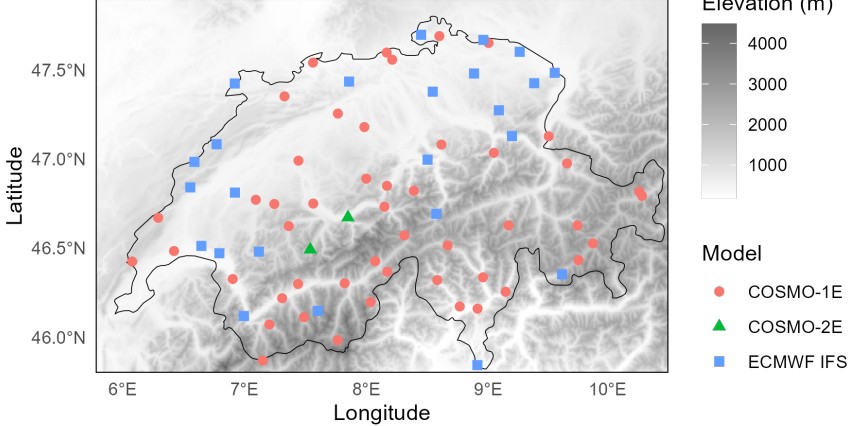

Figure 4: Forecast model at each of the 82 weather stations that receives the highest weight on average. Results are shown at a lead time of 18 hours.

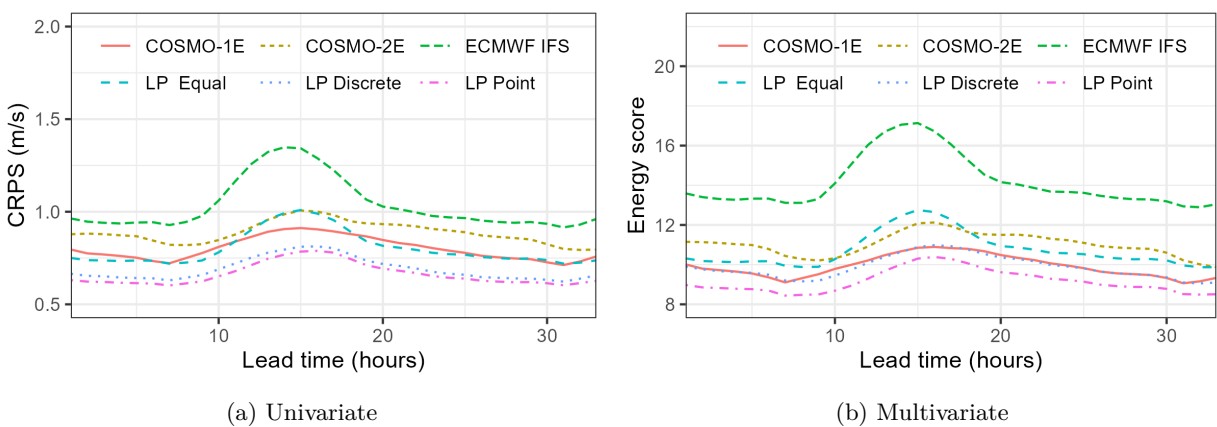

(a) Univariate            (b) Multivariate

Figure 5: Accuracy of the forecasting models and combination methods as a function of lead time. Accuracy is measured using the average CRPS in the univariate case, and the average energy score in the multivariate case. In the univariate case, the scores are averaged across all stations.

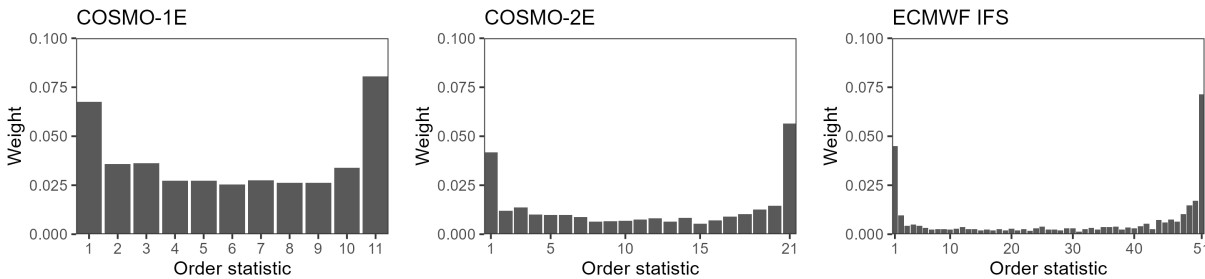

Figure 6: The weight assigned to each order statistic of the samples obtained by the different forecast models.

to post-processed wind speed forecasts; linearly pooling order statistics still results in the most accurate predictions.

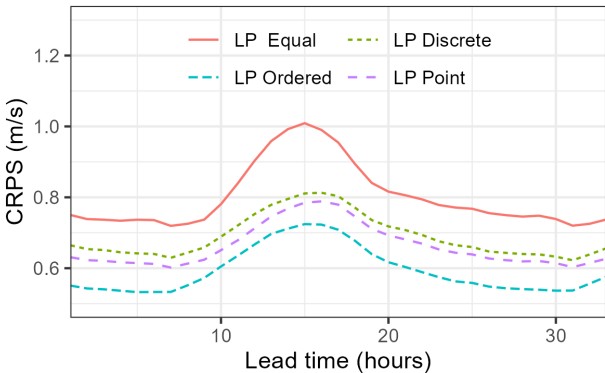

Figure 7: CRPS of the linear pool-based combination methods as a function of lead time. The scores are averaged across all stations.

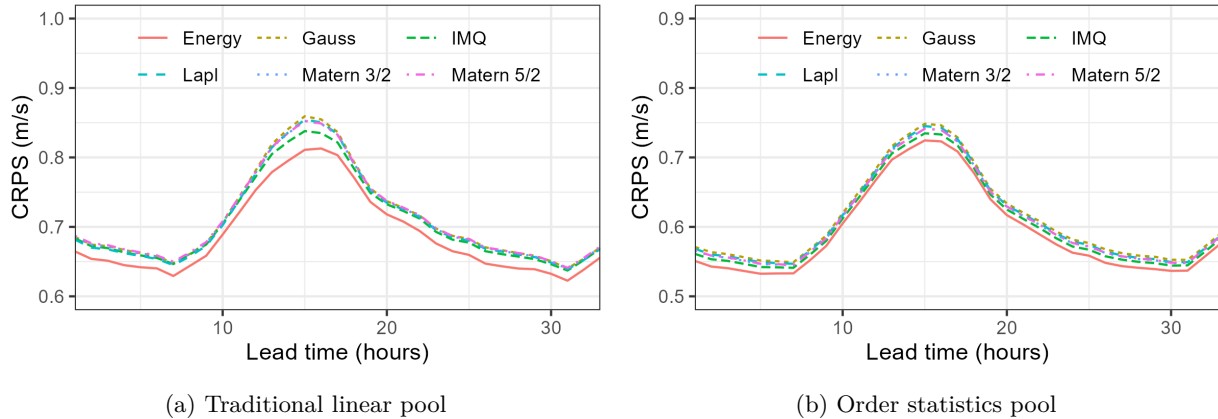

(a) Traditional linear pool

(b) Order statistics pool

Figure 8: CRPS of the linear pool forecast when weights are estimated using various kernel scores, shown as a function of lead time. Results are shown for traditional linear pool and the order statistics pool, for the energy, Gaussian, Laplacian, Matérn 3/2, Matérn 5/2, and inverse multiquadric (IMQ) kernel scores. The scoring rule based on the energy kernel is the CRPS in the univariate case. The scores are averaged across all stations.

## 5.4 Choice of kernel

Thus far, the weights in the pooling strategies have been estimated by optimising the CRPS over the training data. Other kernel scores could be chosen that reward or penalise different aspects of the forecasts, which could result in different behaviour of the estimated weights. However, since weather forecasts are typically evaluated using the CRPS, it also becomes the canonical loss function to employ during model training; estimating the linear pool weights that optimise the CRPS in the training data should in theory result in the lowest CRPS in the test data (assuming a large enough sample size and exchangeability between the training and test data).

To verify this, Figure 8 shows the CRPS of the (univariate) traditional linear pool forecasts and the order statistics pool forecasts when the weights are estimated using kernel scores derived from a range of popular kernels. In addition to the energy kernel, we consider the Gaussian (or squared exponential) kernel, the Laplacian (or exponential) kernel, the Matérn 3/2 and 5/2 kernels, and the inverse multiquadric kernel. These kernels are given in Appendix C. Many of these kernels depend on an additional lengthscale hyperparameter, which is set to one in Figure 8, though similar results are obtained for other lengthscales (see Appendix C).

Unsurprisingly, the most accurate forecasts at all lead times are obtained when the CRPS is used to estimate the weights. The variation in the CRPS between the different kernel choices is on a smaller scale than the variation in the CRPS between the different pooling methods in Figure 5. Nonetheless, if the forecasts were evaluated using a different kernel score, then this kernel score should also be used for weight estimation. Similarly, if multiple kernel scores are to be employed for evaluation, then, since the sum or product of positive definite kernels is itself a positive definite kernel (Rasmussen & Williams, 2006), the kernel score corresponding to this combined kernel could also trivially be employed within the proposed framework.

The estimated weights exhibit very similar behaviour when obtained using the different kernel scores, as shown in Figure 9 in Appendix C. This may partly be because the considered kernels are all stationary; that is, they depend on their arguments $x, x' \in \mathbb{R}$ only via the distance $|x - x'|$. Other non-stationary kernels could also be considered, though the design of such kernel scores is very application specific and has not yet received much attention in the forecasting literature. We therefore refrain from implementing such kernels in this study.

## 6 Efficient pooling on arbitrary outcome domains

In the previous section, the combination strategies are applied univariately, with weights estimated by optimising the CRPS. This is achieved in the RKHS framework of Proposition 2 using the energy kernel with $d = 1$. However, the results of Proposition 2 hold more generally when combining probabilistic predictions made on arbitrary domains. For example, while multivariate combination strategies have received considerably less attention in the literature than univariate approaches, weights corresponding to multivariate predictions could also be estimated efficiently by selecting an appropriate kernel and leveraging the results in Proposition 2.

Consider the weather forecasting application in Section 5. We now show results when the three forecast models are combined with weights estimated by minimising the energy score over the training data. This again corresponds to performing the optimisation within the RKHS of the energy kernel, now with $d = 82$ (the number of stations). In this case, a multivariate forecast can be interpreted as a prediction of the wind speed at all 82 weather stations. The results are qualitatively similar to those when weights are estimated locally, exhibiting similar patterns over lead time and time of year. Figure 3 shows the multivariate weights as a function of lead time. The COSMO-1E forecasts receive yet more weight than in the univariate case, suggesting they also better represent the dependencies in the wind speeds at the different locations. The contributions of COSMO-2E and ECMWF IFS to the linear pool prediction are similar at all lead times.

Figure 2 also shows the weights assigned to each sample member as a function of the multivariate MSE, $\sum_{i=1}^{n} \|x_{j,i}^m - y_i\|^2/n$ for $j = 1, \ldots, J$, and $m = 1, \ldots, M_j$. As in the univariate case, a smaller MSE generally corresponds to a higher weight, with the exception of the control members, which again receive the lowest weight of all sample members.

Unsurprisingly, estimating the weights multivariately results in forecasts that perform worse with respect to the CRPS than when weights are estimated univariately. However, the approach results in valid multivariate predictive distributions, whereas estimating weights separately along each dimension does not. Figure 5 displays the energy score for the different forecasting methods as a function of lead time. The patterns are similar to those observed for the CRPS: COSMO-1E outperforms COSMO-2E and ECMWF IFS at all lead times. Linearly pooling the predictive distributions does not yield large improvements upon the individual component models, instead performing very similarly to COSMO-1E. However, linearly pooling the individual sample members improves energy scores by 5-10% when compared with the traditional linear pool.

As mentioned in Example 6, weighting the order statistics cannot readily be applied beyond the univariate case. To apply the same framework to the multivariate discrete predictive distributions, we must choose an order on $\mathbb{R}^d$ with which to rank the sample members. The choice of this order will result in a different interpretation of the weights, and we therefore leave an analysis of possible orderings for future work.

The results herein could be generalised further to more complex domains. For example, the combination formulas could also be applied to the gridded forecasts prior to interpolating to individual stations, or to the full spatio-temporal trajectories provided by the sample members. This requires the choice of a suitable kernel to employ in the kernel score framework when evaluating these spatio-temporal discrete predictive distributions, the choice of which has not been studied in detail in the forecasting literature. But having selected such a kernel, the combination schemes can be implemented efficiently using the results of Proposition 2.

Furthermore, similarly to as in Section 5.4, while we estimate weights using the energy score, other multivariate kernel scores could similarly be employed for this purpose. Since the energy score is used to assess the multivariate forecasts, it makes sense to use the energy score for weight estimation. Additional results for other kernels are presented in Appendix C.

## 7 Conclusion

In this paper, we demonstrate how well-known results related to positive definite kernels and kernel mean embeddings can be leveraged when combining probabilistic predictions. The standard approach to combine

probabilistic predictions is to linearly pool their distribution functions. Kernel methods provide a means to efficiently estimate the weights in a linear pool by minimising a proper scoring rule over a training data set. The approach embeds the component distributions into an RKHS, and then finds the weights that minimise the maximum mean discrepancy between the linear pool predictions and the corresponding observations. This results in a convex quadratic optimisation problem, facilitating an efficient implementation.

For certain kernels and types of predictive distributions, it may not be possible to derive the kernel mean embeddings analytically. However, this becomes trivial when the predictive distributions are discrete, i.e. the predictions are finite samples. In this case, the approach can readily be applied using any kernel, and in any domain of interest. Through the choice of a suitable kernel, this permits optimal probabilistic combination strategies to be applied not only to predictions for univariate real-valued outcomes, but also to predictions for outcomes on arbitrary domains, including multivariate Euclidean space, spatio-temporal domains, function spaces, and graph spaces.

This also means that, when it is not possible to calculate the kernel mean embeddings for the given kernel and predictive distributions, one could sample from the predictive distributions and then combine the resulting samples using the methods described in Section 6. The sampling could be done randomly, or using evenly-spaced quantiles of the predictive distributions, for example. As the sample size increases, the estimated weights should tend towards the weights that would have been obtained when pooling the original (population level) predictive distributions. For small sample sizes, the estimated weights may change depending on the kernel, the form of the original predictive distributions, and the choice of sampling scheme. Future work could explore this in further detail.

While the linear pool is commonly used to combine probabilistic predictions in practice, the linear pool is somewhat parsimonious, and is known to suffer from some theoretical limitations. Namely, the linear pool does not preserve calibration of the component predictions, instead increasing the dispersion of the predictive distribution (Hora, 2004; Gneiting & Ranjan, 2013). Several non-linear forecast combination strategies have therefore been proposed to circumvent this, generally under the name of the *generalised linear pool* (e.g. Dawid et al., 1995). While these non-linear transformations do not readily align with the RKHS-based estimation framework proposed herein, we study an alternative generalisation of the linear pool that also overcomes some of its theoretical limitations. The approach is capable of quantifying the relative contribution of each component prediction to the combination, whilst simultaneously re-calibrating the combined prediction. In the context of discrete predictive distributions, this corresponds to assigning a separate weight to each order statistic of the samples underlying the predictive distribution, which can be implemented efficiently by embedding the predictions into an RKHS. In an application to operational wind speed forecasts, this approach is found to offer vast improvements upon the traditional linear pool.

Importantly, the proposed approach maintains a straightforward interpretation. In the application to operational weather forecasts, we verify that sample members generated by running numerical weather models from random perturbations of initial conditions appear exchangeable, and find that the control member of a weather forecast (the model run with unperturbed initial conditions) is typically the least informative sample member when constructing a probabilistic prediction, despite it receiving the best mean squared error when compared to the other sample members. We additionally studied the weights assigned to the different weather models at different forecast lead times, times of the day, and altitudes of the weather station, allowing us to understand the relative performance of the component predictions in different circumstances.

However, the results herein are not specific to weather forecasts, and can be used to pool predictions in any domain. Our results in Section 5 focus on weights estimated using the CRPS and energy score, which, while popular in the field of weather and climate forecasting, are rapidly gaining interest in many other domains (Waghmare & Ziegel, 2025). Similarly, discrete predictive distributions are common not only in weather forecasting, but also in energy forecasting (e.g. Berrisch & Ziel, 2023), epidemiological forecasting (e.g. Bracher et al., 2021), and whenever predictive distributions are obtained from generative models or Markov chain Monte-Carlo (MCMC) output (Krüger et al., 2021). Future work could employ the results herein in other application domains.

Combining probabilistic predictions can more generally be interpreted as a distribution-to-distribution regression problem, where the component predictions are distribution-valued covariates, and we are interested

in predicting the conditional distribution of the target variable given these covariates; with it being important that the relative contribution of each covariate to the final prediction can be understood. Linear pooling is one approach, but several other distribution-to-distribution regression methods have been proposed. It would be interesting to study these in the context of probabilistic prediction combination, and to identify whether any can additionally be combined with kernel methods to permit an efficient implementation.

Finally, the proposed framework is one example of how kernel methods can be applied within the context of weather and climate forecasting, and one example of how kernel embeddings can be used to efficiently optimise proper scoring rules. We expect that there are several similar applications of these kernel methods, and we hope that this study motivates their implementation in practice.

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

## A    Efficient pooling of Gaussian predictive distributions

While we focus primarily on discrete forecast distributions, Proposition 2 also holds when linear pooling continuous forecast distributions, provided we can calculate the kernel mean embeddings of the component distributions. To illustrate this, consider the case where $\mathcal{Y} \subseteq \mathbb{R}$, each of the component distributions $F_{j,i}$ is a Gaussian $N(\mu_{j,i}, \sigma_{j,i}^2)$ distribution for $i = 1, \ldots, n$ and $j = 1, \ldots, J$, and we wish to estimate the weights in the linear pool that optimise the CRPS. The linear pool distribution in this case is a Gaussian mixture,

$$F_{LP,i} = \sum_{j=1}^{J} w_j N(\mu_{j,i}, \sigma_{j,i}^2).$$

If $X_{j,i} \sim N(\mu_{j,i}, \sigma_{j,i}^2)$ with $X_{j,i}$ and $X_{l,i}$ independent for $j \neq l$, then

$$\mathbb{E}|X_{j,i} - X_{l,i}| = \sqrt{\sigma_{j,i}^2 + \sigma_{l,i}^2} \sqrt{\frac{2}{\pi}} \exp\left(-\frac{(\mu_{j,i} - \mu_{l,i})^2}{2(\sigma_{j,i} + \sigma_{l,i})^2}\right) + (\mu_{j,i} - \mu_{l,i})\left[1 - 2\Phi\left(-\frac{\mu_{j,i} - \mu_{l,i}}{\sqrt{\sigma_{j,i}^2 + \sigma_{l,i}^2}}\right)\right].$$

The expectation $\mathbb{E}|X_{j,i} - z|$ is given by the same expression with $\mu_{l,i} = z$ and $\sigma_{l,i} = 0$, for any $z \in \mathbb{R}$.

Using this, it becomes straightforward to calculate the expected energy kernel between $X_{j,i}$ and $X_{l,i}$,

$$\mathbb{E}\left[|X_{j,i}| + |X_{l,i}| - |X_{j,i} - X_{l,i}|\right],$$

as well as the expected energy kernel between $X_{j,i}$ and $y_i$,

$$\mathbb{E}\left[|X_{j,i}| + |y_i| - |X_{j,i} - y_i|\right].$$

These can be plugged into equation 10 and equation 11 to get the matrix $A$ and the vector $\boldsymbol{c}$ in equation 12, respectively. The optimal weights in the Gaussian mixture can then be estimated efficiently using existing quadratic programming software, as outlined in Proposition 2. Gaussian mixtures are commonly issued by prediction methods such as Bayesian model averaging (BMA; Raftery et al., 2005), and these results therefore demonstrate that the optimal BMA weights can be estimated efficiently by embedding the forecast distributions into an RKHS.

## B   Continuous weight function estimation

Consider a single probability distribution function $f$ on $\mathcal{Y}$ and a function $w$ from $\mathcal{Y}$ to $[0, \infty)$ such that the integral of the product $wf$ exists and is positive. Up to a renormalisation, $wf$ defines a valid probability density, which we denote by

$$w \square f = \frac{wf}{\int wf}.$$

Such reweightings frequently occur in statistics, for example in importance sampling. Note also that the latter operation is reminiscent of the $\oplus$ operation defined in Bayes Hilbert spaces (van den Boogaart et al., 2014). We now demonstrate how the kernel mean embedding framework of Proposition 2 can be extended to estimate the continuous weight function $w$ that yields the optimum weighted forecast distribution $w \square f$. This can then be further extended to estimate the linear pool of such distributions, as proposed in equation 18.

Suppose that the weight function can be written as a linear combination of basis functions $g_\ell : \mathcal{X} \to [0, \infty)$ for $\ell = 1, \ldots, L$. That is, $w = \sum_{\ell=1}^{L} \beta_\ell g_\ell$, for $\ell_j \geq 0$. For example, the basis functions could be Gaussian distributions, permitting weight functions in the form of Gaussian mixtures. The (renormalised) weighted forecast distribution becomes

$$w \square f = \frac{wf}{\int wf} = \sum_{\ell=1}^{L} \beta_\ell \frac{g_\ell f}{\sum_{j=1}^{L} \beta_j \int g_j f} = \sum_{\ell=1}^{L} \frac{\beta_\ell \int g_\ell f}{\sum_{j=1}^{L} \beta_j \int g_j f}(g_\ell \square f) = \sum_{\ell=1}^{L} \gamma_\ell(g_\ell \square f),$$

where $\gamma_\ell \in [0, 1]$ with $\sum_{\ell=1}^{L} \gamma_\ell = 1$. In other words, $w \square f$ is a mixture of the (renormalised) forecast distribution $f$ weighted by the basis functions $g_\ell$, i.e. $g_\ell \square f$.

In practice, estimating the optimal weighted forecast distribution corresponds to estimating the parameters $\gamma_\ell$ that minimise the empirical score (for a chosen scoring rule) over a set of training data. Since the weight function is a linear combination of the basis functions, $w \square f$ can be interpreted as a linear pool of the basis-weighted forecast densities $g_\ell \square f$, and the problem simplifies to estimating the optimal linear pool weights, $\gamma_\ell$. Using the results from Proposition 2, this is a quadratic problem under linear equality and inequality constraints, which can be practically solved using efficient off-the-shelf implementations. This provides a means to (efficiently) recalibrate predictive densities by assigning more weight to particular regions of the outcome space. The weight assigned to different basis functions can be used to understand how the forecast distribution is altered.

This approach can be extended to estimate continuous weight functions within a linear pool, as in equation 18. Assuming again that the weight function is a linear combination of the basis functions $g_\ell$, we could estimate the weights $\gamma_{j,\ell}$ in

$$f_{OS} = \sum_{j=1}^{J} \sum_{\ell=1}^{L} \gamma_{j,\ell}(g_\ell \square f_j),$$

such that $\gamma_{j,\ell} \in [0,1]$ and $\sum_{j=1}^{J} \sum_{\ell=1}^{L} \gamma_{j,\ell} = 1$. This is similar to the double summation at equation 14 and equation 15, except that the component distributions are continuous distributions rather than Dirac measures. In particular, this can be rewritten as a linear pool of the $J \times L$ forecast distributions $g_\ell \square f_j$, for $j = 1, \ldots, J$ and $\ell = 1, \ldots, L$. The weights that optimise a kernel score can therefore be estimated analogously within the framework of Proposition 2.

## C   Extra case study results

**Choice of kernel**

In Section 5.4, results are studied when different kernel scores are used to estimate the linear pool weights. In addition to the energy kernel, results are presented for the Gaussian (squared exponential) kernel,

$$k_\rho(x, x') = \exp\left(-\frac{\|x - x'\|^2}{\rho}\right);$$

the Laplacian (exponential) kernel,

$$k_\rho(x, x') = \exp\left(-\frac{\|x - x'\|}{\rho}\right);$$

the Matérn 3/2 kernel,

$$k_\rho(x, x') = \left(1 + \frac{\sqrt{3}\|x - x'\|}{\rho}\right)\exp\left(-\frac{\sqrt{3}\|x - x'\|}{\rho}\right);$$

the Matérn 5/2 kernel,

$$k_\rho(x, x') = \left(1 + \frac{\sqrt{5}\|x - x'\|}{\rho} + \frac{5\|x - x'\|^2}{3\rho^2}\right)\exp\left(-\frac{\sqrt{5}\|x - x'\|}{\rho}\right);$$

and the inverse multiquadric (IMQ) kernel,

$$k(x, x') = (1 + \|x - x'\|^2)^{-1/2},$$

for $x, x' \in \mathbb{R}^d$. The Gaussian, Laplace, and Matérn kernels all depend on a lengthscale hyperparameter $\rho > 0$, which is fixed at $\rho = 1$ in Figure 8.

Figure 9 displays the estimated weight assigned to the three forecast models in Section 5 in the (univariate) traditional linear pool. Results are shown when the weight is estimated using the kernels listed above, with three choices of the lengthscale hyperparameter where relevant, $\rho \in \{0.2, 1, 5\}$. The results are largely similar for all kernels: the COSMO-1E model receives the largest weight at all lead times, with the COSMO-2E and ECMWF IFS models assigned similar, smaller weights. This is independent of the family of kernel used, though there are some fluctuations for different choices of the hyperparameters. When $\rho = 0.2$, large errors are penalised more heavily, making overdispersed forecasts preferable to underdispersed forecasts, while the opposite is true for $\rho = 5$. Since the ECMWF IFS forecasts are generally more dispersive than those issued by COSMO-1E and COSMO-2E, this model receives a comparatively higher weight from the Gaussian, Laplacian, and Matérn kernels with $\rho = 0.2$, and a lower weight from these kernels with $\rho = 5$. In this sense, the hyperparameter choice seems to have a larger influence on the estimated linear pool weights than the kernel family, though this may change if non-stationary families of kernels would also be considered.

Figure 10 displays the energy score of the combined forecasts in the multivariate setting of Section 6 when the linear pool weights are estimated using the (multivariate) kernels listed above. Training the combination methods by optimising the energy score generally results in the most accurate forecasts when assessed using the energy score. The traditional linear pool forecasts perform similarly when trained using the inverse multiquadric kernel and the energy kernel. This is likely due to sampling variation, since, unlike in the univariate case, the multivariate scores are not averaged across the different stations.

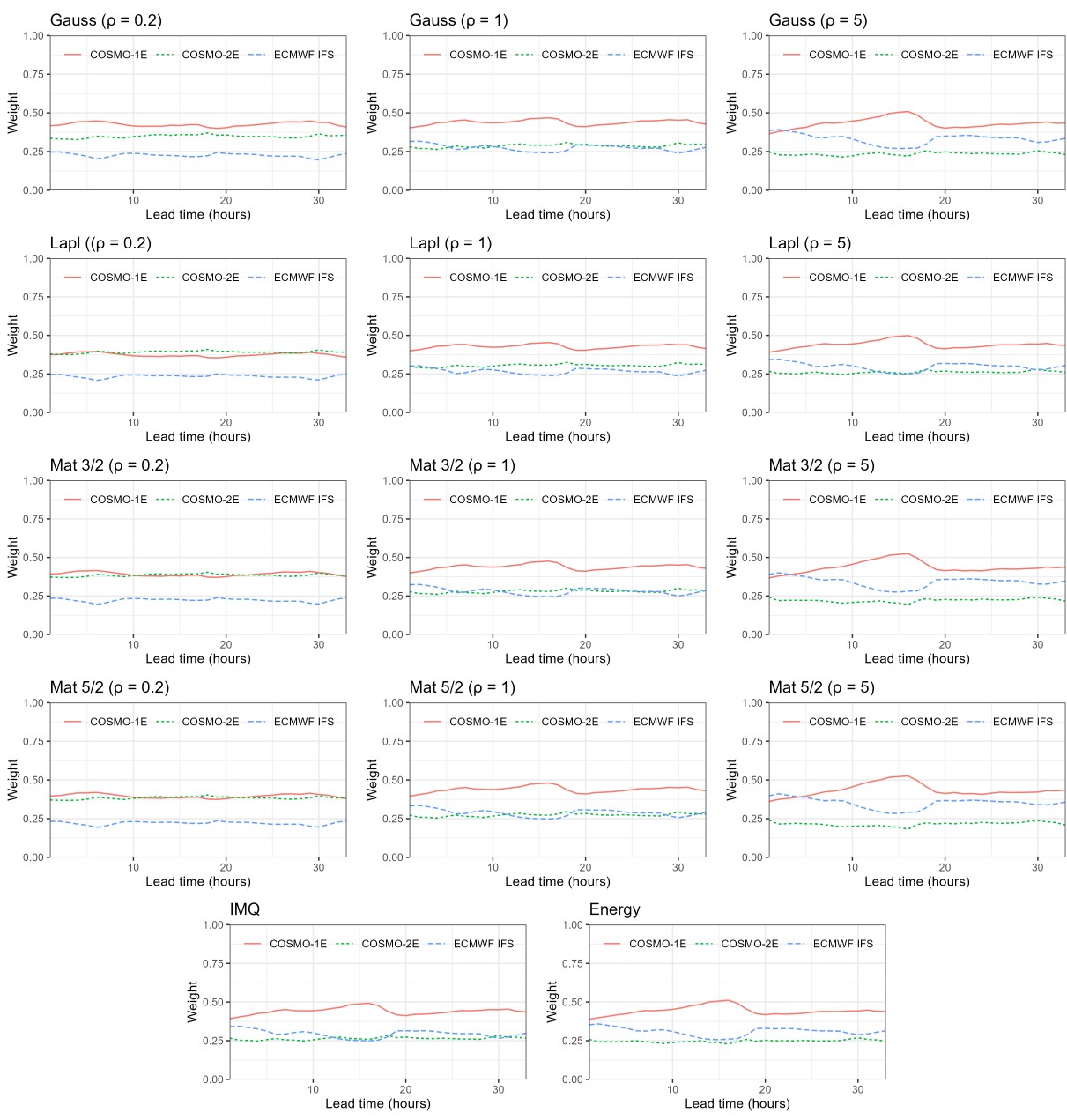

Figure 9: As in Figure 3 but when the weights are estimated using different kernel scores. Results are shown for the Energy, Gaussian, Laplacian, Matérn 3/2, Matérn 5/2, and inverse multiquadric (IMQ) kernels. The Gaussian, Laplacian, and Matérn kernels are shown for lengthscales of $\rho \in \{0.2, 1, 5\}$.

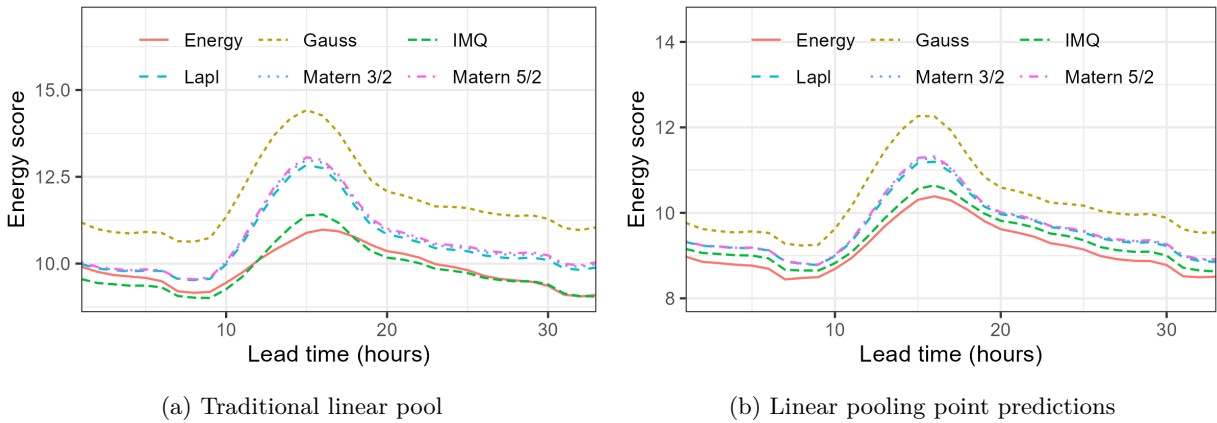

(a) Traditional linear pool

(b) Linear pooling point predictions

Figure 10: Energy score of the linear pool forecast when weights are estimated using various kernel scores, shown as a function of lead time. Results are shown for traditional linear pool and when linearly pooling point predictions, for the energy, Gaussian, Laplacian, Matérn 3/2, Matérn 5/2, and inverse multiquadric (IMQ) kernel scores.

**Post-processed forecasts**

The results in Section 5 correspond to combinations of forecast models that all exhibit large systematic biases. Large improvements can therefore be obtained by implementing an approach that somewhat transforms the predictive distributions to be less biased. In this appendix, we apply the same forecast combination approaches to the forecast models after they have undergone statistical post-processing. Post-processing re-calibrates the output of the numerical weather predictions model, generating predictions that are more reliable. It is therefore well-established within operational weather forecasting suites.

In this case, we implement member-by-member post-processing, which rescales the mean of the discrete predictive distributions, and then shifts the sample members around this mean (Van Schaeybroeck & Vannitsem, 2015). In particular, for each sample $x_{j,i}^1, \ldots, x_{j,i}^{M_j}$, the post-processed discrete predictive distribution contains the transformed members

$$\check{x}_{j,i}^m = (a_j + b_j \bar{x}_{j,i}) + \sqrt{c_j + \frac{d_j}{s_{j,i}^2}}(x_{j,i}^m - \bar{x}_{j,i}),$$

for $m = 1, \ldots, M_j$, where $\bar{x}_{j,i}$ and $s_{j,i}^2$ are the mean and variance, respectively, of the $j$-th discrete predictive distribution, for $i = 1, \ldots, n$, and $a_j, b_j \in \mathbb{R}$, $c_j, d_j \geq 0$ are parameters to be estimated; separate parameters are estimated for each forecast model. The parameters can be estimated easily using the method of moments (Williams, 2016, Chapter 4). Since shifting the sample members can result in negative wind speeds, we apply the transformation to square-root transformed wind speeds, before back transforming to get positive sample members. One desirable property of member-by-member post-processing methods is that they do not change the ordering of the sample members, meaning the multivariate structure contained in the sample members is retained.

The same combination methods can be applied to the post-processed forecasts as were applied in Section 5 to the raw output of the forecast models. Figures 11 to 15 contain the same results as in Section 5 for the post-processed forecasts, while Figures 16 contains probability integral transform (PIT) histograms that assess the calibration of the various prediction strategies.

Figure 11 displays the weight assigned to each of the three post-processed forecasts as a function of lead time. Having removed the biases in the forecasts, the low-resolution ECMWF IFS forecasts are assigned the highest weight. This is reinforced by Figure 12, which shows that ECMWF IFS receives the highest weight at the majority of the weather stations.

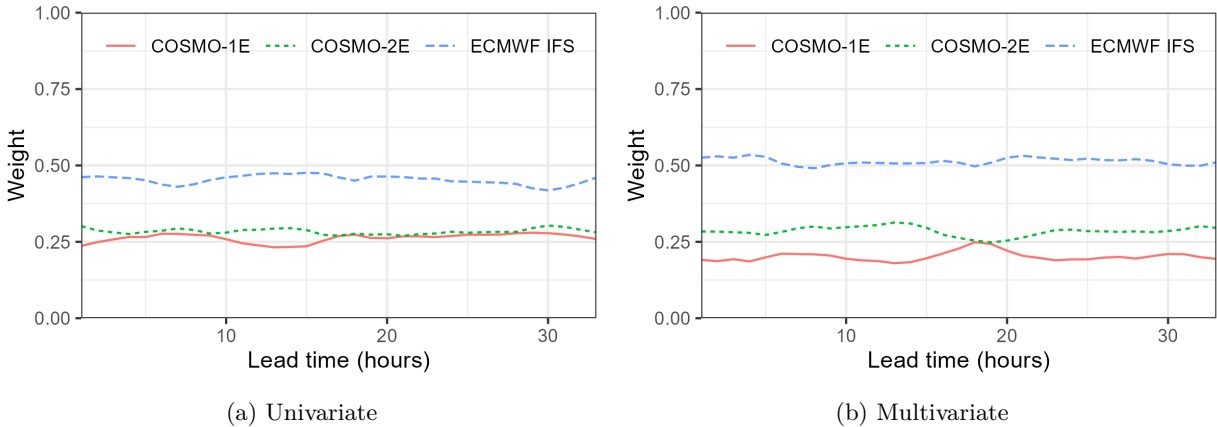

(a) Univariate

(b) Multivariate

Figure 11: As in Figure 3 but for the statistically post-processed predictive distributions.

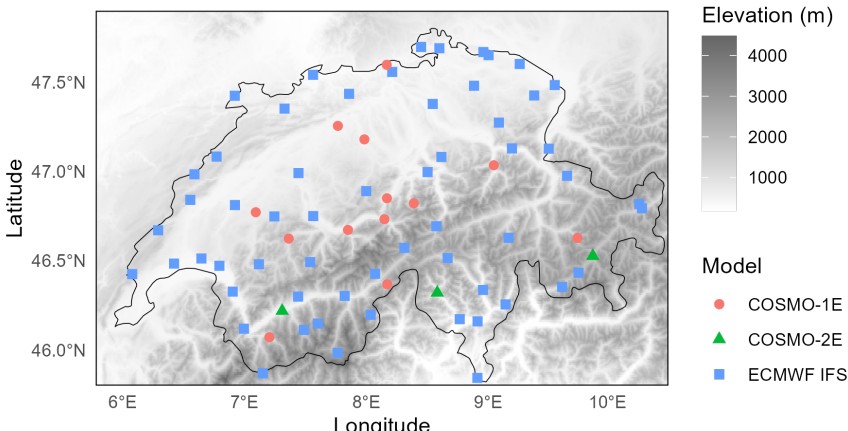

Figure 12: As in Figure 4 but for the statistically post-processed predictive distributions.

From Figure 13, we see that the post-processed COSMO-1E forecasts again outperform those obtained using COSMO-2E and ECMWF IFS at all lead times, suggesting the information contained in the forecasts is retained during post-processing. However, since the post-processed forecasts are considerably more reliable than the raw output from the three forecast models, there is less benefit to the combination strategies. Linearly pooling the sample members of the forecast models marginally outperforms COSMO-1E, the equal-weighted linear pool, and the traditional linear pool at most lead times. However, the post-processed COSMO-1E forecasts are clearly most accurate when assessed multivariately using the energy score.

Figure 14 displays the weights assigned to each order statistic of the discrete predictive distributions when linearly pooling them. In contrast to Figure 6, higher weight is now assigned to the central order statistics. This is likely because linearly pooling calibrated predictions is guaranteed to yield a predictive distribution that is over-dispersive. By assigning a higher weight to the central order statistics, the approach decreases the spread of the component predictive distributions, allowing the resulting combination to be calibrated, as suggested by Proposition 8. This is reinforced by a flatter PIT histogram in Figure 16b than the other combination strategies. Figure 15 demonstrates that this order statistics-based approach yields improvements in accuracy of around 10% upon all other methods at all lead times.

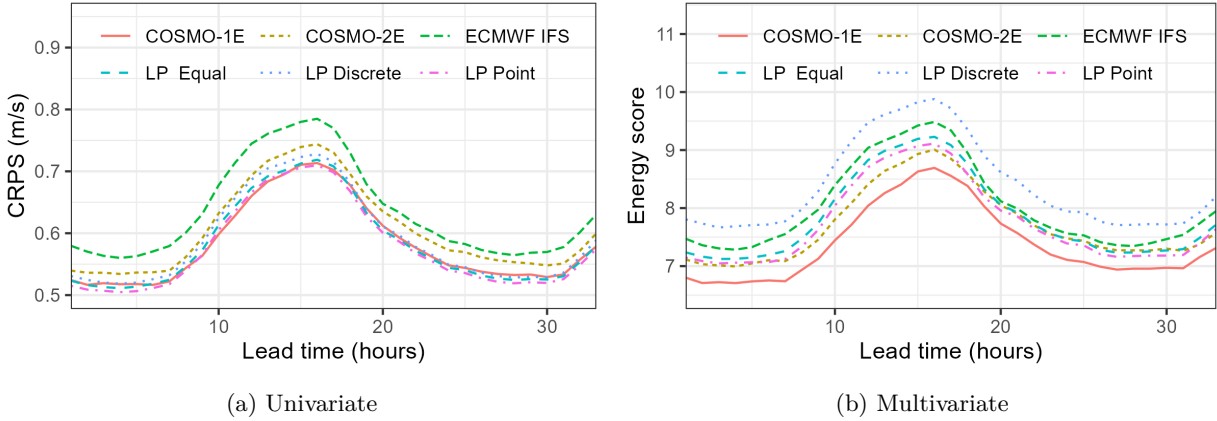

(a) Univariate                                      (b) Multivariate

Figure 13: As in Figure 5 but for the statistically post-processed predictive distributions.

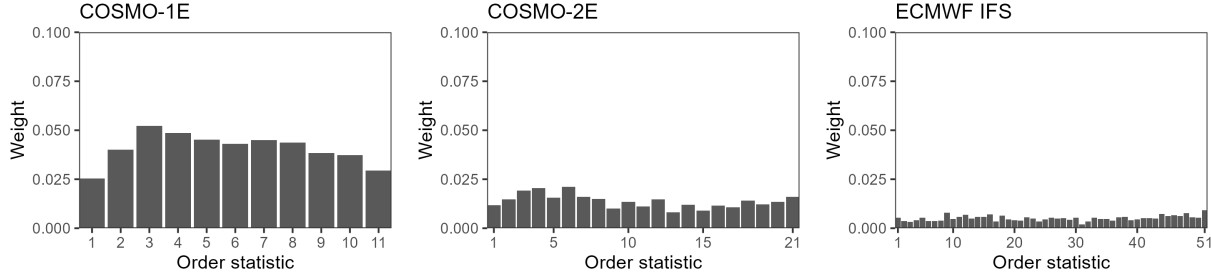

Figure 14: As in Figure 6 but for the statistically post-processed predictive distributions.

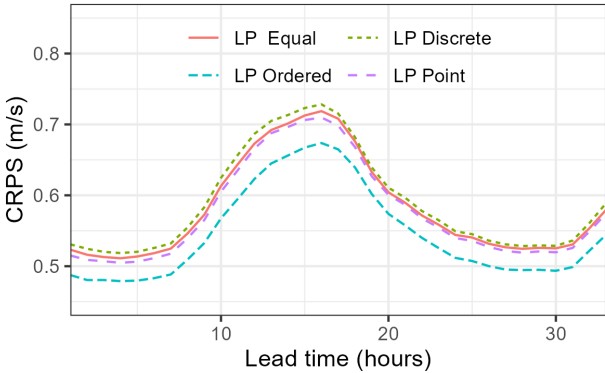

Figure 15: As in Figure 7 but for the statistically post-processed predictive distributions.

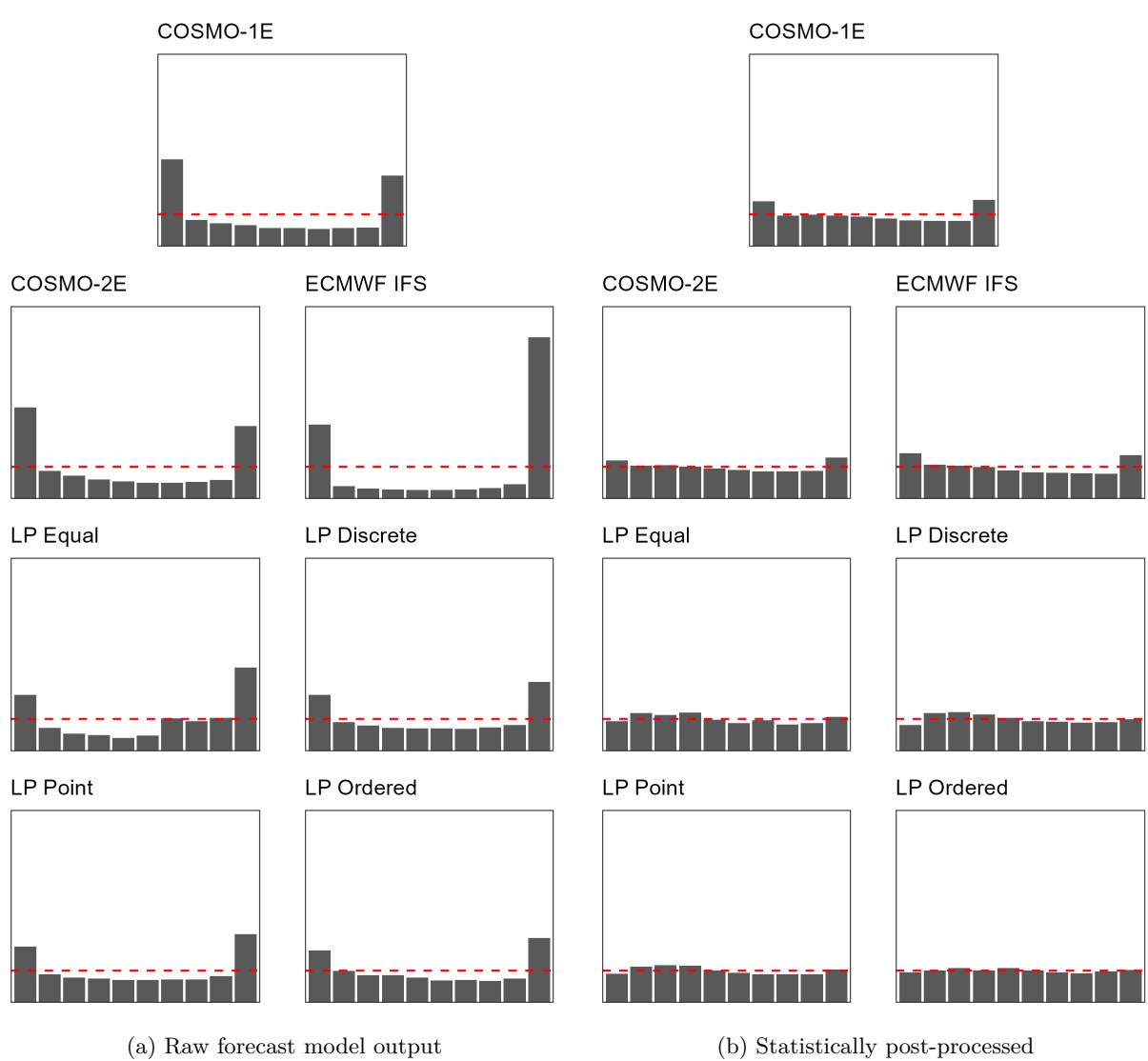

(a) Raw forecast model output                    (b) Statistically post-processed

Figure 16: Probability integral transform (PIT) histograms for the various forecast methods. Results are shown at a lead time of 18 hours, averaged across all stations. The horizontal dashed line has been added at the height of a flat histogram.

