# OpenReview forum: "Efficient pooling of predictions via kernel embeddings"
_TMLR — Accepted by TMLR_

### Review · Reviewer_RojW · 2025-03-01

**Summary Of Contributions:**

This paper studies strategies for pooling of multiple predictions with the aim of producing more skillful and informative forecasts.   The assessment method consider is that of proper scoring rules in the spirit of Dawid and Lauritzen.

Moving beyond the standard equal weighted pooling approach, the authors explore weighted pooling methods,  inititally with constant weights, and subsequently spatially extended weights.     They demonstrate how, in the context of kernel-derived scoring rules, that the optimal weights for a given set of observations, finding the weights which optimise a kernel scoring rule is a convex quadratic programming problem (i.e. quadratic objective, linear constraints).

They then consider functional weights,  noting that his combination strategy can represent arbitrary distributions, as long as the supports are compatible.

They explore this approach on a weather forecasting example - making the very intereresting observation that the higher fidelity forecasts tend to be assigned lower weights - which suggests that these strategies promote uncertainty over tightness.

**Audience:**

Yes

**Claims And Evidence:**

Yes

**Requested Changes:**

* At the very minimum I would encourage a deeper exploration of the interplay between the kernel choice, the dimension and the performance of the linear pool predictor.   Some aspects of this could be readily drawn from existing theory of kernel mean embeddings and MMD.   Others could be explored through a more comprehensive sets of numerical experiments.

* If feasible, I would encourage a more comprehensive exploration of the weight function setting - maybe demonstrating the practical value if this is feasible.    Various works have considered similar functional weights in the context of statistical discrepancies with the objective of promoting robustness - some insights from those contributions might prove useful here.

**Strengths And Weaknesses:**

Strengths:   Generally, the paper is written to a very high standard, and the numerical example considered is both interesting and illuminating.

Weaknesses:
* While I believe the results are of interest and novel, they are quite simple observations, considered in a very simple setting.  Indeed, the main setting under consideration is that of discrete predictive distributions, in which case, the problem reduces to what is effectively finding the weights which minimise the maximum mean discrepancy (MMD) associated with the kernel.
* The setting of functional weights is interesting, but ultimately is not explored in detail - it would have been interesting to understand how this could have been implemented in practice,   how the function weights could be interpreted, etc.
* The choice of kernel itself seems relatively under-explored.   Granted that in the context of environmental forecasting CRPS is widely used, but exploring the influence of the kernel choice would have been interesting.    Besides the (already answered) question of which kernel gives rises to a proper scoring rule, other aspects could be discussed - e.g. the interplay between the optimal weights and the kernel spectrum; the performance of the optimal combination as the dimension goes to infinity, etc.
* In particular, the final part of remark 7: "This generally cannot be solved using the framework of Proposition 2" misses an opportunity in my opinion, in that one could readily add a regularisation term over the w's which should convexify the problem (unless I am mistaken).

---

> ### Author Response · Authors · 2025-05-17
> **Response to reviewer**
>
> Thank you very much for reviewing our manuscript, we have addressed your recommendations in the revised manuscript. The exact amendments that we have made in response to your suggestions are listed below. We look forward to hearing your response.
>
> ## Weaknesses
> [W1] Finding the weights that optimise a kernel score is equivalent to minimising an MMD, for both discrete and continuous predictive distributions. We agree that this is a simple observation for those familiar with kernel methods. However, in the forecasting literature, while the idea to estimate weights or model parameters by optimising a proper scoring rule is well understood, the connection between scoring rules and kernels has not been explored in detail. This paper seeks to highlight how the existing, well-established theory on kernels can be leveraged to help solve common forecasting problems. Hence, while simple, framing linear pool weight estimation in the framework of kernels has not yet been explored, and will likely not be obvious to many practitioners working in the field of forecasting who are unfamiliar with the kernel literature. We have now added some text after Proposition 2 to clarify this.
>
> [W2] The setting of functional weights is not explored in detail since it cannot directly be written as a quadratic optimisation problem, in contrast to in the discrete case. If the weight functions are themselves expressed as a linear combination of basis functions, then the problem becomes linear again in the basis function coefficients. We have added an appendix (Appendix B) that describes how this can be achieved, and have additionally added a paragraph in the main text (Remark 9) to explain this. We choose not to implement this approach in this paper, since the weather forecasts we consider are discrete, and the choice of basis functions is something that should be explored. We therefore leave this for future work.
>
> [W3] We did not explore the choice of kernel in detail, since the kernel used for weight estimation is fixed somewhat by the kernel used to evaluate the forecasts; if a given kernel score is used to evaluate forecasts in the test data set, then it makes sense to find the model parameters that optimise this same kernel score in the training data set (assuming sufficient data is available). We have added a new section (Section 5.4) that verifies this empirically. We employ several popular kernels for weight estimation, including the Gaussian (squared exponential), Laplacian (exponential), and two Matern kernels, and demonstrate that the best performance (as assessed using the CRPS) is obtained when the CRPS is used as the loss function during model training. We have additionally added a subsection in the appendix (now Appendix C) that contains analogous results for weight estimation using different kernels in the multivariate case, as well as figures showing how the estimated weight changes over lead time for the different kernels; the behaviour of the weight is generally insensitive to the chosen kernel, among those considered here.
>
> [W4] A regularisation term over the $w$'s can trivially be added to discourage large weight being assigned to one of the component models; this is true for both the discrete and continuous cases. However, we disagree that the problem of continuous weight estimation can be convexified by adding a regularisation term. Regularisation in the optimisation problem is now discussed in Remark 3 after Proposition 2.
>
> ## Requested changes
>
> [RC1] We now show results in Section 5.4 and Appendix C that demonstrate how the performance of the combined predictions in the weather forecasting application, as well as the behaviour of the estimated weights, depend on the kernel score used for weight estimation. These results are presented for both univariate and multivariate weight estimation.
>
> [RC2] We now explain how the continuous weight function setting could be implemented in the framework of Proposition 2 by representing the weight functions as linear combinations of basis functions. We do not implement this approach in a practical application, since the kernel mean embeddings will depend on both the chosen family of parametric distributions as well as the chosen basis functions, making the results very application specific. We argue that this requires a more comprehensive analysis that goes beyond the scope of this paper.

---

> > ### Comment · Reviewer_RojW · 2025-07-06
> > **Reponse**
> >
> > I have reviewed the updated manuscript and the authors' comments.   Thank you for the clarifications,  I agree with all the points made.    Particuarly the new text in the manuscript has helped clarify many points for me, and I am happy with the changes.

---

### Review · Reviewer_5EJV · 2025-03-03

**Summary Of Contributions:**

This paper investigates pooling prediction for probability ensembles. By drawing connections between CRPS, energy score, and MMD through kernel mean embeddings, the paper shows the optimal pooling weight can be solved through quadratic programming. Then, several special cases for discrete probability predictions are presented. Simulations on weather data shows the potential of this observation, and the advantages of the proposed weighting scheme for ordered statistics.

**Audience:**

Yes

**Claims And Evidence:**

Yes

**Requested Changes:**

Could the author clarify the notation F_{LP,i}? Does this denote the contribution of i-th observation in the pooled distribution? (\mu_{F_{LP}} = \sum \alpha_i \mu_{F_{LP,i})?

**Strengths And Weaknesses:**

Strength:

1. This paper is well written and mostly easy to follow.
2. The formulation of optimal weight assignment into quadratic programming is interesting and makes sense. In fact, similar formulation is quite common in density filtering literature.
3. The simulations are well explained and easy to understand.

Weakness:

The quadratic programming formulation seems to be quite limited to discrete cases, as continuous distribution might result in a weighting function that is difficult to solve, as pointed out by the authors. However, significance is not a must for TMLR submission, so considering the context, this is a minor weakness.

---

> ### Author Response · Authors · 2025-05-17
> **Response to reviewer**
>
> Thank you very much for reviewing our manuscript, we have addressed your recommendations in the revised manuscript. The exact amendments that we have made in response to your suggestions are listed below. We look forward to hearing your response.
>
> ## Weaknesses
> [W1] The quadratic programming formulation also holds for continuous predictive distributions. We now demonstrate how this can be achieved when the component distributions are all normal distributions, and the weights are to be optimised using the CRPS. This is discussed in a new appendix (Appendix A). The extension to estimating continuous weight functions, as outlined in Section 4.2, becomes difficult in this case, but can be achieved within the same framework if the weight function is expressed as a linear combination of basis functions. This is also presented in a new appendix (Appendix B).
>
> ## Requested changes
>
> [RC1] The index $i$ corresponds to the indexing over the forecast cases that comprise the training data; this will often be an index over time, for example. Hence, $F_{LP,i}$ corresponds to the linear pool of $F_{1, i}, \dots, F_{J, i}$, the component forecasts of the $i$-th training data case. This has now been clarified in the text (in the beginning of the proof of Proposition 2).

---

> > ### Comment · Reviewer_5EJV · 2025-05-29
> > **Response to rebuttal**
> >
> > Thank you for the response and it clears up my questions.

---

### Review · Reviewer_qWWh · 2025-05-24

**Summary Of Contributions:**

This paper shows how to combine probabilistic predictions using kernel embeddings. It proves that optimizing the weights for linear pooling under kernel scoring rules leads to a convex quadratic problem. This makes the process fast and efficient. The authors also generalize the method to allow weights that vary across the outcome space. The method works on many types of outcome spaces. The paper includes a case study on weather forecasts, showing that this new method outperforms traditional pooling approaches.

**Audience:**

Yes

**Broader Impact Concerns:**

No concerns.

**Claims And Evidence:**

Yes

**Requested Changes:**

* While the main theoretical result (Proposition 2) uses expectations to build the optimization objective, the following implementations use sampled points. Thus, there is a gap between theory and practice, and it is worth discussing how this would impact performance.

**Strengths And Weaknesses:**

Strengths:
* The method is mathematically sound and efficient.
* The generalization to order statistic pooling is novel and performs well.
* The case study is detailed and demonstrates real-world impact.

Weaknesses:
* Some results are application-specific (weather forecasts) and may not generalize easily.

---

> ### Author Response · Authors · 2025-05-26
> **Response to reviewer**
>
> Thank you very much for reviewing our manuscript, we have addressed your recommendations in the revised manuscript. The exact amendments that we have made in response to your suggestions are listed below. We look forward to hearing your response.
>
> ## Weaknesses
> [W1] The theoretical results hold when linearly pooling arbitrary probabilistic predictions, making them relevant for any domain in which predictions are pooled. This includes weather and climate forecasting, but also domains such as energy, finance, medicine, and economics. The application in Section 5 focuses on weight estimation using the CRPS and energy score. While these scoring rules are very popular in the field of weather and climate forecasting, they are rapidly gaining interest in many other domains. Similarly, discrete predictive distributions are common not only in weather forecasting, but also in energy forecasting, epidemiological forecasting, and whenever predictive distributions are obtained from generative models or Markov chain Monte-Carlo (MCMC) output. We have now added a paragraph to the conclusions (Section 6) that justifies this in further detail.
>
> ## Requested changes
> [RC1] The analysis in Section 4 is a specific application of Proposition 2 to discrete predictive distributions, in which case expectations are equal to (weighted) sums over the support points of the distributions. We agree that if the kernel mean embedding for a given kernel and predictive distribution cannot be calculated explicitly, then one could also sample from the predictive distribution and then apply the results in our Section 4 for discrete predictive distributions. However, this is not what we consider in Section 4, in that the weights would be estimates of the weights assigned to the original predictive distributions, rather than weights assigned to each of the discrete predictive distributions. Nonetheless, for large sample sizes, this should still be a feasible weight estimation approach. This is discussed in more detail in a new paragraph in Section 6.

---

### Decision · Action_Editor_bYnK · 2025-08-10

**Recommendation:** Accept as is

**Audience:**

Yes

**Audience Explanation:**

This paper will be of interest to researchers working in probabilistic forecast and associated evaluations based on kernel scoring rules.

**Claims And Evidence:**

Yes

**Claims Explanation:**

The paper addresses the problem of linear pooling of multiple predictions by finding the combining weights that optimise a proper scoring rule over some training data. Through the connections with kernel scoring rules and kernel mean embedding, the paper shows that estimating the linear pool weights that optimise kernel-based scoring rules is a convex quadratic optimisation problem (Proposition 2). The focus of this work is on finite distributions, but the revised manuscript also discusses how to extend the proposed method to continuous distributions, thereby broadening the scope of this work. The paper is well written, the theoretical results are sound, and the empirical results, while limited to weather forecasts, demonstrate the benefits of the proposed method.

All expert reviewers lean toward acceptance, advocating that the paper is well written and will be of interest to communities doing work in probabilistic forecasting and associated evaluation. Based on my own evaluation of the paper, I concur with the reviewers' evaluation. The authors have addressed all of the concerns raised by the reviewers in the revision.